# MultiGuard: Provably Robust Multi-label Classification against Adversarial Examples

**Jinyuan Jia**[*]
University of Illinois Urbana-Champaign
jinyuan@illinois.edu

**Wenjie Qu**[*]
Huazhong University of Science and Technology
wen_jie_qu@outlook.com

**Neil Zhenqiang Gong**
Duke University
neil.gong@duke.edu

## Abstract

Multi-label classification, which predicts a set of labels for an input, has many applications. However, multiple recent studies showed that multi-label classification is vulnerable to adversarial examples. In particular, an attacker can manipulate the labels predicted by a multi-label classifier for an input via adding carefully crafted, human-imperceptible perturbation to it. Existing provable defenses for multi-class classification achieve sub-optimal provable robustness guarantees when generalized to multi-label classification. In this work, we propose MultiGuard, the first provably robust defense against adversarial examples to multi-label classification. Our MultiGuard leverages randomized smoothing, which is the state-of-the-art technique to build provably robust classifiers. Specifically, given an arbitrary multi-label classifier, our MultiGuard builds a smoothed multi-label classifier via adding random noise to the input. We consider isotropic Gaussian noise in this work. Our major theoretical contribution is that we show a certain number of ground truth labels of an input are provably in the set of labels predicted by our MultiGuard when the $\ell_2$-norm of the adversarial perturbation added to the input is bounded. Moreover, we design an algorithm to compute our provable robustness guarantees. Empirically, we evaluate our MultiGuard on VOC 2007, MS-COCO, and NUS-WIDE benchmark datasets. Our code is available at: https://github.com/quwenjie/MultiGuard

## 1 Introduction

*Multi-class classification* assumes each input only has one ground truth label and thus often predicts a single label for an input. In contrast, in *multi-label classification* [42, 41, 35, 43], each input has multiple ground truth labels and thus a multi-label classifier predicts a set of labels for an input. For instance, an image could have multiple objects, attributes, or scenes. Multi-label classification has many applications such as diseases detection [16], object recognition [43], retail checkout recognition [18], document classification [34], etc..

However, similar to multi-class classification, multiple recent studies [56, 53, 30] showed that multi-label classification is also vulnerable to adversarial examples. In particular, an attacker can manipulate the set of labels predicted by a multi-label classifier for an input via adding carefully crafted perturbation to it. Adversarial examples pose severe security threats to the applications of multi-label classification in security-critical domains. To mitigate adversarial examples to multi-label

---

[*]Equal contribution. Wenjie Qu performed this research when he was a remote intern in Gong's group.

36th Conference on Neural Information Processing Systems (NeurIPS 2022).

classification, several *empirical defenses* [49, 1, 30] have been proposed. For instance, Melacci et al. [30] proposed to use the domain knowledge on the relationships among the classes to improve the robustness of multi-label classification. However, these defenses have no provable robustness guarantees, and thus they are often broken by more advanced attacks. For instance, Melacci et al. [30] showed that their proposed defense can be broken by an adaptive attack that exploits the domain knowledge used in the defense. Moreover, existing provably robust defenses [10, 8, 17, 46, 12, 22] are all for multi-class classification, which achieve sub-optimal provable robustness guarantee when extended to multi-label classification as shown by our experimental results.

**Our work:** We propose *MultiGuard*, the first provably robust defense against adversarial examples for multi-label classification. MultiGuard leverages randomized smoothing [5, 29, 24, 26, 12], which is the state-of-the-art technique to build provably robust classifiers. In particular, compared to other provably robust techniques, randomized smoothing has two advantages: 1) scalable to large-scale neural networks, and 2) applicable to any classifiers. Suppose we have an arbitrary multi-label classifier (we call it *base multi-label classifier*), which predicts $k'$ labels for an input. We build a *smoothed multi-label classifier* via randomizing an input. Specifically, given an input, we first create a *randomized input* via adding random noise to it. We consider the random noise to be isotropic Gaussian in this work. Then, we use the base multi-label classifier to predict labels for the randomized input. Due to the randomness in the randomized input, the $k'$ labels predicted by the base multi-label classifier are also random. We use $p_i$ to denote the probability that the label $i$ is among the set of $k'$ labels predicted by the base multi-label classifier for the randomized input, where $i \in \{1, 2, \cdots, c\}$. We call $p_i$ *label probability*. Our smoothed multi-label classifier predicts the $k$ labels with the largest label probabilities for the input. We note that $k'$ and $k$ are two different parameters.

Our main theoretical contribution is to show that, given a set of labels (e.g., the ground truth labels) for an input, at least $e$ of them are provably in the set of $k$ labels predicted by MultiGuard for the input, when the $\ell_2$-norm of the adversarial perturbation added to the input is no larger than a threshold. We call $e$ *certified intersection size*. We aim to derive the certified intersection size for MultiGuard. However, existing randomized smoothing studies [24, 12, 22] achieves sub-optimal provable robustness guarantees when generalized to derive our certified intersection size. The key reason is they were designed for multi-class classification instead of multi-label classification. Specifically, they can guarantee that a smoothed multi-class classifier provably predicts the same single label for an input [24, 12] or a certain label is provably among the top-$k$ labels predicted by the smoothed multi-class classifier [22]. In contrast, our certified intersection size characterizes the intersection between the set of ground truth labels of an input and the set of labels predicted by a smoothed multi-label classifier. In fact, previous provable robustness results [12, 22] are special cases of ours, e.g., our results reduce to Cohen et al. [12] when $k' = k = 1$ and Jia et al. [22] when $k' = 1$.

In particular, there are two challenges in deriving the certified intersection size. The first challenge is that the base multi-label classifier predicts multiple labels for an input. The second challenge is that an input has multiple ground truth labels. To solve the first challenge, we propose a variant of Neyman-Pearson Lemma [33] that is applicable to multiple functions, which correspond to multiple labels predicted by the base multi-label classifier. In contrast, existing randomized smoothing studies [24, 26, 12, 22] for multi-class classification use the standard Neyman-Pearson Lemma [33] that is only applicable for a single function, since their base multi-class classifier predicts a single label for an input. To address the second challenge, we propose to use the *law of contraposition* to simultaneously consider multiple ground truth labels of an input when deriving the certified intersection size.

Our derived certified intersection size is the optimal solution to an optimization problem, which involves the label probabilities. However, it is very challenging to compute the exact label probabilities due to the continuity of the isotropic Gaussian noise and the complexity of the base multi-label classifiers (e.g., complex deep neural networks). In response, we design a Monte Carlo algorithm to estimate the lower or upper bounds of label probabilities with probabilistic guarantees. More specifically, we can view the estimation of lower or upper bounds of label probabilities as a binomial proportion confidence interval estimation problem in statistics. Therefore, we use the Clopper-Pearson [11] method from the statistics community to obtain the label probability bounds. Given the estimated lower or upper bounds of label probabilities, we design an efficient algorithm to solve the optimization problem to obtain the certified intersection size.

Empirically, we evaluate our MultiGuard on VOC 2007, MS-COCO, and NUS-WIDE benchmark datasets. We use the *certified top-k precision@R*, *certified top-k recall@R*, and *certified top-k f1-score@R* to evaluate our MultiGuard. Roughly speaking, certified top-$k$ precision@$R$ is the least fraction of the $k$ predicted labels that are ground truth labels of an input when the $\ell_2$-norm of the adversarial perturbation is at most $R$; certified top-$k$ recall@$R$ is the least fraction of ground truth labels of an input that are in the set of $k$ labels predicted by our MultiGuard; and certified top-$k$ f1-score@$R$ is the harmonic mean of certified top-$k$ precision@$R$ and certified top-$k$ recall@$R$. Our experimental results show that our MultiGuard outperforms the state-of-the-art certified defense [22] when extending it to multi-label classification. For instance, on VOC 2007 dataset, Jia et al. [22] and our MultiGuard respectively achieve 24.3% and 31.3% certified top-$k$ precision@$R$, 51.6% and 66.4% certified top-$k$ recall@$R$, as well as 33.0% and 42.6% certified top-$k$ f1-score@$R$ when $k' = 1$, $k = 3$, and $R = 0.5$.

Our major contributions can be summarized as follows:

- We propose MultiGuard, the first provably robust defense against adversarial examples for multi-label classification.
- We design a Monte Carlo algorithm to compute the certified intersection size.
- We evaluate our MultiGuard on VOC 2007, MS-COCO, and NUS-WIDE benchmark datasets.

## 2   Background and Related Work

**Multi-label classification:** In multi-label classification, a multi-label classifier predicts multiple labels for an input. Many deep learning classifiers [27, 52, 43, 45, 57, 32, 21, 7, 54, 48, 2, 13] have been proposed for multi-label classification. For instance, a naive method for multi-label classification is to train independent binary classifiers for each label and use ranking or thresholding to derive the final predicted labels. This method, however, ignores the topology structure among labels and thus cannot capture the label co-occurrence dependency (e.g., *mouse* and *keyboard* usually appear together). In response, several methods [43, 7] have been proposed to improve the performance of multi-label classification via exploiting the label dependencies in an input. Despite their effectiveness, these methods rely on complicated architecture modifications. To mitigate the issue, some recent studies [48, 2] proposed to design new loss functions. For instance, Baruch et al. [2] introduced an asymmetric loss (ASL). Roughly speaking, their method is based on the observation that, in multi-label classification, most inputs contain only a small fraction of the possible candidate labels, which leads to under-emphasizing gradients from positive labels during training. Their experimental results indicate that their method achieves state-of-the-art performance on multiple benchmark datasets.

**Adversarial examples to multi-label classification:** Several recent studies [40, 56, 53, 30, 20] showed that multi-label classification is vulnerable to adversarial examples. An attacker can manipulate the set of labels predicted by a multi-label classifier for an input via adding carefully crafted perturbation to it. For instance, Song et al. [40] proposed white-box, targeted attacks to multi-label classification. In particular, they first formulate their attacks as optimization problems and then use gradient descent to solve them. Their experimental results indicate that they can make a multi-label classifier produce an arbitrary set of labels for an input via adding adversarial perturbation to it. Yang et al. [53] explored the worst-case mis-classification risk of a multi-label classifier. In particular, they formulate the problem as a bi-level set function optimization problem and leverage random greedy search to find an approximate solution. Zhou et al. [56] proposed to generate $\ell_\infty$-norm adversarial perturbations to fool a multi-label classifier. In particular, they transform the optimization problem of finding adversarial perturbations into a linear programming problem which can be solved efficiently.

**Existing empirically robust defenses:** Some studies [49, 1, 30] developed empirical defenses to mitigate adversarial examples in multi-label classification. For instance, Wu et al. [49] applied adversarial training, a method developed to train robust multi-class classifiers, to improve the robustness of multi-label classifiers. Melacci et al. [30] showed that domain knowledge, which measures the relationships among classes, can be used to detect adversarial examples and improve the robustness of multi-label classifiers. However, all these defenses lack provable robustness guarantees and thus, they are often broken by advanced adaptive attacks. For instance, Melacci et al. [30] showed that their defenses can be broken by adaptive attacks that also consider the domain knowledge.

**Existing provably robust defenses:** All existing provably robust defenses [37, 10, 6, 19, 8, 17, 46, 4, 24, 12, 26, 36, 23, 47, 39, 31, 44, 38, 55, 50] were designed for multi-class classification instead of multi-label classification. In particular, they can guarantee that a robust multi-class classifier

predicts the same single label for an input or a label (e.g., the single ground truth label of the input) is among the top-$k$ labels predicted by a robust multi-class classifier. These defenses are sub-optimal for multi-label classification. Specifically, in multi-label classification, we aim to guarantee that at least some ground truth labels of an input are in the set of labels predicted by a robust multi-label classifier.

MultiGuard leverages randomized smoothing [24, 26, 12, 22, 51]. Existing randomized smoothing studies (e.g., Jia et al. [22]) achieve sub-optimal provable robustness guarantees (i.e., certified intersection size) for multi-label classification, because they are designed for multi-class classification. For example, as our empirical evaluation results will show, MultiGuard significantly outperforms Jia et al. [22] when extending it to multi-label classification. Technically speaking, our work has two key differences with Jia et al.. First, the base multi-class classifier in Jia et al. only predicts a single label for an input while our base multi-label classifier predicts multiple labels for an input. Second, Jia et al. can only guarantee that a single label is provably among the $k$ labels predicted by a smoothed multi-class classifier, while we aim to show that multiple labels (e.g., ground truth labels of an input) are provably among the $k$ labels predicted by a smoothed multi-label classifier. Due to such key differences, we require new techniques to derive the certified intersection size of MultiGuard. For instance, we develop a variant of Neyman-Pearson Lemma [33] which is applicable to multiple functions while Jia et al. uses the standard Neyman-Pearson Lemma [33] which is only applicable to a single function. Moreover, we use the law of contraposition to derive our certified intersection size, which is not required by Jia et al..

## 3   Our MultiGuard

### 3.1   Building our MultiGuard

**Label probability:**  Suppose we have a multi-label classifier $f$ which we call *base multi-label classifier*. Given an input $\mathbf{x}$, the base multi-label classifier $f$ predicts $k'$ labels for it. For simplicity, we use $f_{k'}(\mathbf{x})$ to denote the set of $k'$ labels predicted by $f$ for $\mathbf{x}$. We use $\epsilon$ to denote an isotropic Gaussian noise, i.e., $\epsilon \sim \mathcal{N}(0, \sigma^2 \cdot I)$, where $\sigma$ is the *standard deviation* and $I$ is an *identity matrix*. Given $\mathbf{x} + \epsilon$ as input, the output of $f$ would be random due to the randomness of $\epsilon$, i.e., $f_{k'}(\mathbf{x} + \epsilon)$ is a random set of $k'$ labels. We define *label probability* $p_i$ as the probability that the label $i$ is among the set of top-$k'$ labels predicted by $f$ when adding isotropic Gaussian noise to an input $\mathbf{x}$, where $i \in \{1, 2, \cdots, c\}$. Formally, we have $p_i = \Pr(i \in f_{k'}(\mathbf{x} + \epsilon))$.

**Our smoothed multi-label classifier:**  Given the label probability $p_i$'s for an input $\mathbf{x}$, our *smoothed multi-label classifier* $g$ predicts the $k$ labels with the largest label probabilities for $\mathbf{x}$. For simplicity, we use $g_k(\mathbf{x})$ to denote the set of $k$ labels predicted by our smoothed multi-label classifier for an input $\mathbf{x}$.

**Certified intersection size:**  An attacker adds a perturbation $\delta$ to an input $\mathbf{x}$. $g_k(\mathbf{x} + \delta)$ is the set of $k$ labels predicted by our smoothed multi-label classifier for the perturbed input $\mathbf{x} + \delta$. Given a set of labels $L(\mathbf{x})$ (e.g., the ground truth labels of $\mathbf{x}$), our goal is to show that at least $e$ of them are in the set of $k$ labels predicted by our smoothed multi-label classifier for the perturbed input, when the $\ell_2$-norm of the adversarial perturbation is at most $R$. Formally, we aim to show the following:

$$\min_{\delta, \|\delta\|_2 \leq R} |L(\mathbf{x}) \cap g_k(\mathbf{x} + \delta)| \geq e, \tag{1}$$

where we call $e$ *certified intersection size*. Note that different inputs may have different certified intersection sizes.

### 3.2   Deriving the Certified Intersection Size

**Defining two random variables:**  Given an input $\mathbf{x}$, we define two random variables $\mathbf{X} = \mathbf{x} + \epsilon$, $\mathbf{Y} = \mathbf{x} + \delta + \epsilon$, where $\delta$ is an adversarial perturbation and $\epsilon$ is isotropic Gaussian noise. Roughly speaking, the random variables $\mathbf{X}$ and $\mathbf{Y}$ respectively denote the inputs derived by adding isotropic Gaussian noise to the input $\mathbf{x}$ and its adversarially perturbed version $\mathbf{x} + \delta$. Based on the definition of the label probability, we have $p_i = \Pr(i \in f_{k'}(\mathbf{X}))$. We define *adversarial label probability* $p_i^*$ as $p_i^* = \Pr(i \in f_{k'}(\mathbf{Y})), i \in \{1, 2, \cdots, c\}$. Intuitively, adversarial label probability $p_i^*$ is the probability that the label $i$ is in the set of $k'$ labels predicted by the base multi-label classifier $f$ for $\mathbf{Y}$. Given an adversarially perturbed input $\mathbf{x} + \delta$, our smoothed multi-label classifier predicts the $k$ labels with the largest adversarial label probabilities $p_i^*$'s for it.

**Derivation sketch:** We leverage the *law of contraposition* in our derivation. Roughly speaking, if we have a statement: $P \longrightarrow Q$, then its contrapositive is: $\neg Q \longrightarrow \neg P$, where $\neg$ is the logical negation symbol. The law of contraposition claims that a statement is true if and only if its contrapositive is true. In particular, we define the following predicate:

$$Q : \min_{\delta, \|\delta\|_2 \leq R} |L(\mathbf{x}) \cap g_k(\mathbf{x} + \delta)| \geq e. \tag{2}$$

Intuitively, $Q$ is true if at least $e$ labels in $L(\mathbf{x})$ can be found in $g_k(\mathbf{x} + \delta)$ for an arbitrary adversarial perturbation $\delta$ whose $\ell_2$-norm is no larger than $R$. Then, we have $\neg Q : \min_{\delta, \|\delta\|_2 \leq R} |L(\mathbf{x}) \cap g_k(\mathbf{x} + \delta)| < e$. Moreover, we derive a necessary condition (denoted as $\neg P$) for $\neg Q$ to be true, i.e., $\neg Q \longrightarrow \neg P$. Roughly speaking, $\neg P$ compares upper bounds of the adversarial label probabilities of the labels in $\{1, 2, \cdots, c\} \setminus L(\mathbf{x})$ with lower bounds of those in $L(\mathbf{x})$. More specifically, $\neg P$ represents that the lower bound of the $e$th largest adversarial label probability of labels in $L(\mathbf{x})$ is no larger than the upper bound of the $(k - e + 1)$th largest adversarial label probability of the labels in $\{1, 2, \cdots, c\} \setminus L(\mathbf{x})$. Finally, based on the law of contraposition, we have $P \longrightarrow Q$, i.e., $Q$ is true if $P$ is true (i.e., $\neg P$ is false).

The major challenges we face when deriving the necessary condition $\neg P$ are as follows: (1) the adversarial perturbation $\delta$ can be arbitrary as long as its $\ell_2$-norm is no larger than $R$, which has infinitely many values, and (2) the complexity of the classifier (e.g., a complex deep neural network) and the continuity of the random variable $\mathbf{Y}$ make it hard to compute the adversarial label probabilities. We propose an innovative method to solve the challenges based on two key observations: (1) the random variable $\mathbf{Y}$ reduces to $\mathbf{X}$ under no attacks (i.e., $\delta = \mathbf{0}$) and (2) the adversarial perturbation $\delta$ is bounded, i.e., $\|\delta\|_2 \leq R$. Our core idea is to bound the adversarial label probabilities using the label probabilities. Suppose we have the following bounds for the label probabilities (we propose an algorithm to estimate such bounds in Section 3.3):

$$p_i \geq \underline{p_i}, \forall i \in L(\mathbf{x}), \tag{3}$$
$$p_j \leq \overline{p}_j, \forall j \in \{1, 2, \cdots, c\} \setminus L(\mathbf{x}). \tag{4}$$

Given the bounds for label probabilities, we derive a lower bound of the adversarial label probability for each label $i \in L(\mathbf{x})$ and an upper bound of the adversarial label probability for each label $j \in \{1, 2, \cdots, c\} \setminus L(\mathbf{x})$. To derive these bounds, we propose a variant of the Neyman-Pearson Lemma [33] which enables us to consider multiple functions. In contrast, the standard Neyman-Pearson Lemma [33] is insufficient as it is only applicable to a single function while the base multi-label classifier outputs multiple labels.

We give an overview of our derivation of the bounds of the adversarial label probabilities and show the details in the proof of the Theorem 1 in supplementary material. Our idea is to construct some regions in the domain space of $\mathbf{X}$ and $\mathbf{Y}$ via our variant of the Neyman-Pearson Lemma. Specifically, given the constructed regions, we can obtain the lower/upper bounds of the adversarial label probabilities using the probabilities that the random variable $\mathbf{Y}$ is in these regions. Note that the probabilities that the random variables $\mathbf{X}$ and $\mathbf{Y}$ are in these regions can be easily computed as we know their probability density functions.

Next, we derive a lower bound of the adversarial label probability $p_i^*$ ($i \in L(\mathbf{x})$) as an example to illustrate our main idea. Our derivation of the upper bound of the adversarial label probability for a label in $\{1, 2, \cdots, c\} \setminus L(\mathbf{x})$ follows a similar procedure. Given a label $i \in L(\mathbf{x})$, we can find a region $\mathcal{A}_i$ via our variant of Neyman-Pearson Lemma [33] such that $\Pr(\mathbf{X} \in \mathcal{A}_i) = \underline{p_i}$. Then, we can derive a lower bound of $p_i^*$ via computing the probability of the random variable $\mathbf{Y}$ in the region $\mathcal{A}_i$, i.e., we have:

$$p_i^* \geq \Pr(\mathbf{Y} \in \mathcal{A}_i). \tag{5}$$

The above lower bound can be further improved via jointly considering multiple labels in $L(\mathbf{x})$. Suppose we use $\Gamma_u \subseteq L(\mathbf{x})$ to denote an arbitrary set of $u$ labels. We can craft a region $\mathcal{A}_{\Gamma_u}$ via our variant of Neyman-Pearson Lemma such that we have $\Pr(\mathbf{X} \in \mathcal{A}_{\Gamma_u}) = \frac{\sum_{i \in \Gamma_u} \underline{p_i}}{k'}$. Then, we can derive the following lower bound:

$$\max_{i \in \Gamma_u} p_i^* \geq \frac{k'}{u} \cdot \Pr(\mathbf{Y} \in \mathcal{A}_{\Gamma_u}). \tag{6}$$

The $e$th largest lower bounds of adversarial label probabilities of labels in $L(\mathbf{x})$ can be derived by combing the lower bounds in Equation 5 and 6. Formally, we have the following theorem:

**Theorem 1** (Certified Intersection Size). *Suppose we are given an input* $\mathbf{x}$*, a base multi-label classifier* $f$*, our smoothed classifier* $g$*, and a set of* $d$ *ground truth labels* $L(\mathbf{x}) = \{a_1, a_2, \cdots, a_d\}$ *for* $\mathbf{x}$*. Moreover, we have a lower bound* $\underline{p_i}$ *of* $p_i$ *for each* $i \in L(\mathbf{x})$ *satisfying Equation 3 and an upper bound* $\overline{p_j}$ *of* $p_j$ *for each* $j \in \{1, 2, \cdots, c\} \setminus L(\mathbf{x})$ *satisfying Equation 4. We assume* $\underline{p_{a_1}} \geq \cdots \geq \underline{p_{a_d}}$ *for convenience. Let* $\overline{p_{b_1}} \geq \overline{p_{b_2}} \geq \cdots \geq \overline{p_{b_{c-d}}}$ *be the* $c - d$ *label probability upper bounds for the labels in* $\{1, 2, \cdots, c\} \setminus L(\mathbf{x})$*, where ties are broken uniformly at random. Given a perturbation size* $R$*, we have the following guarantee:*

$$\min_{\delta, \|\delta\|_2 \leq R} |L(\mathbf{x}) \cap g_k(\mathbf{x} + \delta)| \geq e, \tag{7}$$

*where* $e$ *is the optimal solution to the following optimization problem or 0 if it does not have a solution:*

$$e = \underset{e'=1,2,\cdots,\min\{d,k\}}{\operatorname{argmax}} e'$$

$$s.t. \max\{\Phi(\Phi^{-1}(\underline{p_{a_{e'}}}) - \frac{R}{\sigma}), \max_{u=1}^{\eta} \frac{k'}{u} \cdot \Phi(\Phi^{-1}(\frac{\underline{p_{A_u}}}{k'}) - \frac{R}{\sigma})\}$$

$$> \min\{\Phi(\Phi^{-1}(\overline{p_{b_s}}) + \frac{R}{\sigma}), \min_{v=1}^{s} \frac{k'}{v} \cdot \Phi(\Phi^{-1}(\frac{\overline{p_{B_v}}}{k'}) + \frac{R}{\sigma})\}, \tag{8}$$

*where* $\Phi$ *and* $\Phi^{-1}$ *respectively are the cumulative distribution function and its inverse of the standard Gaussian distribution,* $\eta = d - e' + 1$*,* $\underline{p_{A_u}} = \sum_{l=e'}^{e'+u-1} \underline{p_{a_l}}$*,* $s = k - e' + 1$*, and* $\overline{p_{B_v}} = \sum_{l=s-v+1}^{s} \overline{p_{b_l}}$*.*

*Proof.* Please refer to Appendix A in supplementary material. $\qquad\square$

We have the following remarks for our theorem:

- When $k' = k = 1$ and $L(\mathbf{x})$ only contains a single label, our certified intersection size reduces to the robustness result derived by Cohen et al. [12], i.e., the smoothed classifier provably predicts the same label for an input when the adversarial perturbation is bounded. When $k' = 1$, $k \geq 1$, and $L(\mathbf{x})$ only contains a single label, our certified intersection size reduces to the robustness result derived by Jia et al. [22], i.e., a label is provably among the $k$ labels predicted by a smoothed classifier when the adversarial perturbation is bounded. In other words, the certified robustness guarantees derived by Cohen et al. [12] and Jia et al. [22] are special cases of our results. Note that Cohen et al. is a special case of Jia et al. Moreover, both Cohen et al. and Jia et al. focused on certifying robustness for multi-class classification instead of multi-label classification.

- Our certified intersection size holds for arbitrary attacks as long as the $\ell_2$-norm of the adversarial perturbation is no larger than $R$. Moreover, our results are applicable for any base multi-label classifier.

- Our certified intersection size relies on a lower bound of the label probability for each label $i \in L(\mathbf{x})$ and an upper bound of the label probability for each label $j \in \{1, 2, \cdots, c\} \setminus L(\mathbf{x})$. Moreover, when the label probability bounds are estimated more accurately, our certified intersection size may be larger.

- Our theorem requires $\underline{p_{A_u}} \leq k'$ and $\overline{p_{B_v}} \leq k'$. We have $\underline{p_{A_u}} \leq p_{A_u} \leq \sum_{i \in L(\mathbf{x})} p_i \leq \sum_{j=1}^{c} p_j = k'$. In Section 3.3, our estimated $\overline{p_{B_v}}$ will always be no larger than $k'$. Thus, we can apply our theorem in practice.

- We note that there are respectively two terms in the left- and right-hand sides of Equation 8. The major technical challenge in our derivation stems from the second term in each side. As we will show in our experiments, those two terms significantly improve certified intersection size.

### 3.3 Computing the Certified Intersection Size

In order to compute the certified intersection size for an input $\mathbf{x}$, we need to solve the optimization problem in Equation 8. The key challenge of solving the optimization problem is to estimate lower bounds $\underline{p_i}$ of label probabilities for $i \in L(\mathbf{x})$ and upper bounds $\overline{p_j}$ of label probabilities for $j \in \{1, 2, \cdots, c\} \setminus L(\mathbf{x})$. To address the challenge, we design a Monte Carlo algorithm to estimate these label probability bounds with probabilistic guarantees. Then, given the estimated label probability bounds, we solve the optimization problem to obtain the certified intersection size.

**Estimating label probability bounds:** We randomly sample $n$ Gaussian noise from $\epsilon$ and add them to the input $\mathbf{x}$. We use $\mathbf{x}^1, \mathbf{x}^2, \cdots, \mathbf{x}^n$ to denote the $n$ noisy inputs for convenience. Given these noisy inputs, we use the base multi-label classifier $f$ to predict $k'$ labels for each of them. Moreover, we define the *label frequency* $n_i$ of label $i$ as the number of noisy inputs whose predicted $k'$ labels include $i$. Formally, we have $n_i = \sum_{t=1}^{n} \mathbb{I}(i \in f_{k'}(\mathbf{x}^t)), i \in \{1, 2, \cdots, c\}$, where $\mathbb{I}$ is an indicator function. Based on the definition of label probability $p_i$, we know that $n_i$ follows a *binomial distribution* with parameters $n$ and $p_i$, where $n$ is the number of noisy inputs and $p_i$ is the label probability of label $i$. Our goal is to estimate a lower or upper bound of $p_i$ based on $n_i$ and $n$, which is a binomial proportion confidence interval estimation problem. Therefore, we can use the Clopper-Pearson [11] method from the statistics community to estimate these label probability bounds. Formally, we have the following label probability bounds:

$$\underline{p_i} = \text{Beta}(\frac{\alpha}{c}; n_i, n - n_i + 1), i \in L(\mathbf{x}), \tag{9}$$

$$\overline{p}_j = \text{Beta}(1 - \frac{\alpha}{c}; n_j, n - n_j + 1), \forall j \in \{1, 2, \cdots, c\} \setminus L(\mathbf{x}), \tag{10}$$

where $1 - \frac{\alpha}{c}$ is the confidence level and $\text{Beta}(\rho; \varsigma, \vartheta)$ is the $\rho$th quantile of the Beta distribution with shape parameters $\varsigma$ and $\vartheta$. Based on *Bonferroni correction* [3, 14], the overall confidence level for the $c$ label probability upper or lower bounds is $1 - \alpha$. To solve the optimization problem in Equation 8, we also need to estimate $\underline{p}_{A_u}$ and $\overline{p}_{B_v}$. In particular, we can estimate $\underline{p}_{A_u} = \sum_{l=e'}^{e'+u-1} \underline{p}_{a_l}$ and $\overline{p}_{B_v} = \sum_{l=s-v+1}^{s} \overline{p}_{b_l}$. However, this bound may be loose for $\overline{p}_{B_v}$. We can further improve the bound via considering the constraint that $\overline{p}_{B_v} + \sum_{i \in L(\mathbf{x})} \underline{p_i} \leq k'$. In other words, we have $\overline{p}_{B_v} \leq k' - \sum_{i \in L(\mathbf{x})} \underline{p_i}$. Given this constraint, we can estimate $\overline{p}_{B_v} = \min(\sum_{l=s-v+1}^{s} \overline{p}_{b_l}, k' - \sum_{i \in L(\mathbf{x})} \underline{p_i})$. Note that the above constraint is not applicable for $\underline{p}_{A_u}$ since it is a lower bound.

**Solving the optimization problem in Equation 8:** Given the estimated label probability lower or upper bounds, we solve the optimization problem in Equation 8 via binary search.

**Complete algorithm:** Algorithm 1 in supplementary materials shows our complete algorithm to compute the certified intersection size for an input $\mathbf{x}$. The function RANDOMSAMPLE returns $n$ noisy inputs via first sampling $n$ noise from the isotropic Gaussian distribution and then adding them to the input $\mathbf{x}$. Given the label frequency for each label and the overall confidence $\alpha$ as input, the function PROBBOUNDESTIMATION aims to estimate the label probability bounds based on Equation 9 and 10. The function BINARYSEARCH returns the certified intersection size via solving the optimization problem in Equation 8 using binary search.

# 4 Evaluation

## 4.1 Experimental Setup

**Datasets:** We adopt the following multi-label classification benchmark datasets:

- **VOC 2007 [15]:** Pascal Visual Object Classes Challenge (VOC 2007) dataset [15] contains 9,963 images from 20 objects (i.e., classes). On average, each image has 2.5 objects. Following previous work [43], we split the dataset into 5,011 training images and 4,952 testing images.
- **MS-COCO [28]:** Microsoft-COCO (MS-COCO) [28] dataset contains 82,081 training images, 40,504 validation images, and 40,775 testing images from 80 objects. Each image has 2.9 objects on average. The images in the testing dataset do not have ground truth labels. Therefore, following previous work [7], we evaluate our method on the validation dataset.
- **NUS-WIDE [9]:** NUS-WIDE dataset [9] originally contains 269,648 images from Flickr. The images are manually annotated into 81 visual concepts, with 2.4 visual concepts per image on average. Since the URLs of certain images are not accessible, we adopt the version released by [2], which contains 154,000 training images and 66,000 testing images.

Similar to previous work [12, 25] on certified defenses for multi-class classification, we randomly sample 500 images from the testing (or validation) dataset of each dataset to evaluate our MultiGuard.

**Base multi-label classifiers:** We adopt ASL [2] to train the base multi-label classifiers on the three benchmark datasets. In particular, ASL leverages an asymmetric loss to solve the positive-negative imbalance issue (an image has a few positive labels while has many negative labels on average) in multi-label classification, and achieves state-of-the-art performance on the three benchmark datasets. Suppose $q_j$ is the probability that a base multi-label classifier predicts label $j$ ($j = 1, 2, \cdots, c$) for a

training input. Moreover, we let $y_j$ be 1 (or 0) if the label $j$ is (or is not) a ground truth label of the training input. The loss of ASL [2] is as follows: $L_{ASL} = \sum_{j=1}^{c} -y_j L_{j+} - (1 - y_j) L_{j-}$, where $L_{j+} = (1 - q_j)^{\gamma_+} \log(q_j)$ and $L_{j-} = (\max(q_j - m, 0))^{\gamma_-} \log(1 - \max(q_j - m, 0))$. Note that $\gamma_+, \gamma_-$, and $m$ are hyperparameters. Following [2], we set training hyperameters $\gamma_+ = 0, \gamma_- = 4$, and $m = 0.05$. We train the classifier using Adam optimizer, using learning rate $10^{-3}$ and batch size 32. We adopt the public implementation of ASL[2] in our experiments. Similar to previous work [12] on randomized smoothing based multi-class classification, we add isotropic Gaussian noise to the training data when we train our base multi-label classifiers. In particular, given a batch of training images, we add isotropic Gaussian noise to each of them, and then we use the noisy training images to update the base multi-label classifier. Our experimental results indicate that such training method can substantially improve the robustness of our MultiGuard (please refer to Figure 6 in supplementary material).

**Evaluation metrics:** We use *certified top-k precision@R*, *certified top-k recall@R*, and *certified top-k f1-score@R* as evaluation metrics. We first define them for a single testing input and then for multiple testing inputs. In particular, given the certified intersection size $e$ for a testing input $\mathbf{x}$ under perturbation size $R$, we define them as follows: certified top-k precision@$R = e/k$, certified top-k recall@$R = e/|L(\mathbf{x})|$, certified top-k f1-score@$R = 2 \cdot e/(|L(\mathbf{x})| + k)$, where $L(\mathbf{x})$ is the set of ground truth labels of $\mathbf{x}$ and the symbol $|\cdot|$ measures the number of elements in a set. Roughly speaking, certified top-k precision@$R$ is the *least* fraction of the $k$ predicted labels for an input that are ground truth labels, when the $\ell_2$-norm of the adversarial perturbation is at most $R$; certified top-k recall@$R$ is the least fraction of the ground truth labels in $L(\mathbf{x})$ that are in the $k$ labels predicted by MultiGuard; and certified top-k f1-score@$R$ measures a trade-off between certified top-k precision@$R$ and certified top-k recall@$R$. Note that the above definition is for a single testing input. Given a testing/validation dataset with multiple testing inputs, the overall certified top-k precision@$R$, certified top-k recall@$R$, and certified top-k f1-score@$R$ are computed as the averages over the testing inputs.

**Compared methods:** We compare with the state-of-the-art certified defense, namely Jia et al. [22], by extending it to multi-label classification. To the best of our knowledge, Jia et al. is the only certified defense that considers top-k predictions against $\ell_2$-norm adversarial perturbations. Given an arbitrary label, Jia et al. derived a certified radius such that the label is among the top-k labels predicted by the smoothed classifier (Theorem 1 in Jia et al.). For each label in $L(\mathbf{x})$, we compute a certified radius. Given a perturbation size $R$, the certified intersection size $e$ can be computed as the number of labels in $L(\mathbf{x})$ whose certified radii are larger than $R$.

**Parameter setting:** Our MultiGuard has the following parameters: $k'$ (the number of labels predicted by the base multi-label classifier for an input), $k$ (the number of labels predicted by our smoothed multi-label classifier for an input), standard deviation $\sigma$, number of noisy inputs $n$, and confidence level $1 - \alpha$. Unless otherwise mentioned, we adopt the following default parameters: $\alpha = 0.001$, $n = 1,000$, $\sigma = 0.5$, $k' = 1$ and $k = 3$ for VOC 2007 dataset, and $k' = 3$ and $k = 10$ for MS-COCO and NUS-WIDE datasets, where we use larger $k'$ and $k$ for MS-COCO and NUS-WIDE because they have more classes.

### 4.2 Experimental Results

**Comparison results:** The first row in Figure 1 shows the comparison results on VOC 2007 dataset in default setting. We find that MultiGuard achieves higher certified top-k precision@$R$, certified top-k recall@$R$, and certified top-k f1-score@$R$ than Jia et al.. MultiGuard is better than Jia et al. because MultiGuard jointly considers all ground truth labels, while Jia et al. can only consider each ground truth label independently. For instance, suppose we have two ground truth labels; it is very likely that both of them are not in the top-k predicted labels when considered independently, but at least one of them is among the top-k predicted labels when considered jointly. The intuition is that it is easier for an attacker to find an adversarial perturbation such that a certain label is not in the top-k predicted labels, but it is more challenging for an attacker to find an adversarial perturbation such that both of the two labels are not in the top-k predicted labels. Our observations on the other two datasets are similar, which can be found in Figure 2 in supplementary material.

---

[2]https://github.com/Alibaba-MIIL/ASL

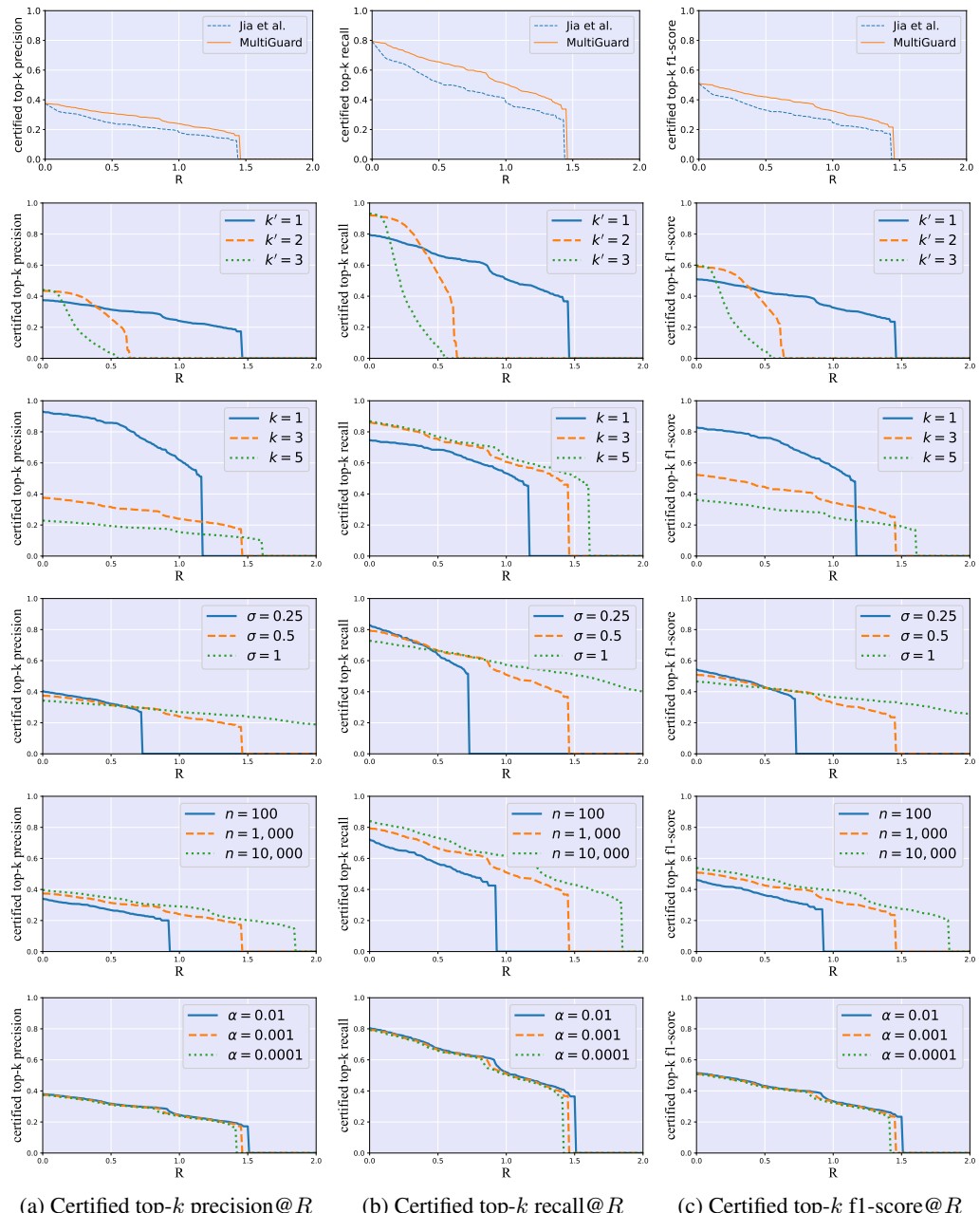

(a) Certified top-$k$ precision@$R$     (b) Certified top-$k$ recall@$R$     (c) Certified top-$k$ f1-score@$R$

Figure 1: Comparing with Jia et al. [22] (first row). Impact of $k'$ (second row), $k$ (third row), $\sigma$ (fourth row), $n$ (fifth row), and $\alpha$ (sixth row) on certified top-$k$ precision@$R$, certified top-$k$ recall@$R$, and certified top-$k$ f1-score@$R$. The dataset is VOC 2007. The results on the other two datasets are shown in supplementary material.

**Impact of $k'$:** The second row in Figure 1 shows the impact of $k'$ on VOC 2007 dataset. In particular, we find that a larger $k'$ achieves a larger certified top-$k$ precision@$R$ (or certified top-$k$ recall@$R$ or certified top-$k$ f1-score@$R$) without attacks (i.e., $R = 0$), but the curves drop more quickly as $R$ increases (i.e., a larger $k'$ is less robust against adversarial examples as $R$ increases). The reason is that a larger $k'$ gives an attacker a larger attack space. Note that this is also reflected in our optimization problem in Equation 8. In particular, the left-hand (or right-hand) side in Equation 8 decreases (or increases) as $k'$ increases, which leads to smaller certified intersection size $e$ as $k'$ increases. We have similar observations on MS-COCO and NUS-WIDE datasets. Please refer to Figure 3 in supplementary material.

**Impact of $k$:** The third row in Figure 1 shows the impact of $k$ on VOC 2007 dataset. We have the following observations from our experimental results. First, $k$ achieves a tradeoff between the certified top-$k$ precision@$R$ without attacks and robustness. In particular, a larger $k$ gives us a smaller certified top-$k$ precision@$R$ without attacks, but the curve drops more slowly as $R$ increases (i.e., a larger $k$ is more robust against adversarial examples as $R$ increases). Second, we find that certified top-$k$ recall@$R$ increases as $k$ increases. The reason is that more labels are predicted by our MultiGuard as $k$ increases. Third, similar to certified top-$k$ precision@$R$, $k$ also achieves a tradeoff between the certified top-$k$ f1-score@$R$ without attacks and robustness. We also have those observations on the other two datasets. Please refer to Figure 4 in supplementary material for details.

**Impact of $\sigma$:** The fourth row in Figure 1 shows the impact of $\sigma$ on VOC 2007 dataset. The experimental results indicate that $\sigma$ achieves a tradeoff between the certified top-$k$ precision@$R$ (or certified top-$k$ recall@$R$ or certified top-$k$ f1-score@$R$) without attacks (i.e., $R = 0$) and robustness. Specifically, a larger $\sigma$ leads to a smaller certified top-$k$ precision@$R$ (or certified top-$k$ recall@$R$ or certified top-$k$ f1-score@$R$) without attacks, but is more robust against adversarial examples as $R$ increases. The observations on the other two datasets are similar. Please refer to Figure 5 in supplementary material.

**Impact of $n$ and $\alpha$:** The fifth and sixth rows in Figure 1 respectively show the impact of $n$ and $\alpha$ on VOC 2007 dataset. In particular, we find that certified top-$k$ precision@$R$ (or certified top-$k$ recall@$R$ or certified top-$k$ f1-score@$R$) increases as $n$ or $\alpha$ increases. The reason is that a larger $n$ or $\alpha$ gives us tighter lower or upper bounds of the label probabilities, which leads to larger certified intersection sizes. However, we find that the certified top-$k$ precision@$R$ (or certified top-$k$ recall@$R$ or certified top-$k$ f1-score@$R$) is insensitive to $\alpha$ once it is small enough and insensitive to $n$ once it is large enough.

**Effectiveness of the second terms in Equation 8:** We perform experiments under our default setting to validate the effectiveness of our second terms in the left- and right-hand sides of Equation 8. Our results are as follows: with and without the second terms, MultiGuard respectively achieves 31.3% and 23.6% certified top-$k$ precision@$R$, 66.4% and 48.8% certified top-$k$ recall@$R$, as well as 42.6% and 31.8% certified top-$k$ f1-score@$R$, where the perturbation size $R = 0.5$ and the dataset is VOC 2007. As the result shows, our second terms can significantly improve certified intersection size.

## 5   Conclusion

In this paper, we propose MultiGuard, the first provably robust defense against adversarial examples for multi-label classification. In particular, we show that a certain number of ground truth labels of an input are provably predicted by our MultiGuard when the $\ell_2$-norm of the adversarial perturbation added to the input is bounded. Moreover, we design an algorithm to compute the certified robustness guarantees. Empirically, we conduct experiments on VOC 2007, MS-COCO, and NUS-WIDE benchmark datasets to validate our MultiGuard. Interesting future work to improve MultiGuard includes: 1) incorporating the knowledge of the base multi-label classifier, and 2) designing new methods to train more accurate base multi-label classifiers.

**Acknowledgements:**   We thank the anonymous reviewers for constructive comments. This work was supported by NSF under Grant No. 1937786 and 2125977 and the Army Research Office under Grant No. W911NF2110182.

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
