# A Proof of Theorem 1

We first define some notations that will be used in our proof. Given an input $\mathbf{x}$, we define the following two random variables:

$$\mathbf{X} = \mathbf{x} + \epsilon \sim \mathcal{N}(\mathbf{x}, \sigma^2 I), \tag{11}$$

$$\mathbf{Y} = \mathbf{x} + \delta + \epsilon \sim \mathcal{N}(\mathbf{x} + \delta, \sigma^2 I), \tag{12}$$

where $\epsilon \sim \mathcal{N}(0, \sigma^2 I)$ and $\delta$ is an adversarial perturbation that has the same size with $\mathbf{x}$. The random variables $\mathbf{X}$ and $\mathbf{Y}$ represent random inputs obtained by adding isotropic Gaussian noise to the input $\mathbf{x}$ and its perturbed version $\mathbf{x} + \delta$, respectively. Cohen et al. [12] applied the standard Neyman-Pearson lemma [33] to the above two random variables, and obtained the following two lemmas:

**Lemma 1** (Neyman-Pearson lemma for Gaussian with different means). *Let* $\mathbf{X} \sim \mathcal{N}(\mathbf{x}, \sigma^2 I)$, $\mathbf{Y} \sim \mathcal{N}(\mathbf{x} + \delta, \sigma^2 I)$, *and* $F : \mathbb{R}^d \to \{0, 1\}$ *be a random or deterministic function. Then, we have the following:*

*(1) If* $W = \{\mathbf{w} \in \mathbb{R}^d : \delta^T \mathbf{w} \leq \beta\}$ *for some* $\beta$ *and* $Pr(F(\mathbf{X}) = 1) \geq Pr(\mathbf{X} \in W)$, *then* $Pr(F(\mathbf{Y}) = 1) \geq Pr(\mathbf{Y} \in W)$.

*(2) If* $W = \{\mathbf{w} \in \mathbb{R}^d : \delta^T \mathbf{w} \geq \beta\}$ *for some* $\beta$ *and* $Pr(F(\mathbf{X}) = 1) \leq Pr(\mathbf{X} \in W)$, *then* $Pr(F(\mathbf{Y}) = 1) \leq Pr(\mathbf{Y} \in W)$.

**Lemma 2.** *Given an input* $\mathbf{x}$, *a real number* $q \in [0, 1]$, *as well as regions* $\mathcal{A}$ *and* $\mathcal{B}$ *defined as follows:*

$$\mathcal{A} = \{\mathbf{w} : \delta^T(\mathbf{w} - \mathbf{x}) \leq \sigma \|\delta\|_2 \, \Phi^{-1}(q)\}, \tag{13}$$

$$\mathcal{B} = \{\mathbf{w} : \delta^T(\mathbf{w} - \mathbf{x}) \geq \sigma \|\delta\|_2 \, \Phi^{-1}(1 - q)\}, \tag{14}$$

*we have the following:*

$$Pr(\mathbf{X} \in \mathcal{A}) = q, \tag{15}$$

$$Pr(\mathbf{X} \in \mathcal{B}) = q, \tag{16}$$

$$Pr(\mathbf{Y} \in \mathcal{A}) = \Phi(\Phi^{-1}(q) - \frac{\|\delta\|_2}{\sigma}), \tag{17}$$

$$Pr(\mathbf{Y} \in \mathcal{B}) = \Phi(\Phi^{-1}(q) + \frac{\|\delta\|_2}{\sigma}). \tag{18}$$

*Proof.* Please refer to [12]. $\square$

Next, we first generalize the Neyman-Pearson lemma to the case of multiple functions and then derive the lemmas that will be used in our proof.

**Lemma 3.** *Let* $\mathbf{X}$, $\mathbf{Y}$ *be two random variables whose probability densities are respectively* $Pr(\mathbf{X} = \mathbf{w})$ *and* $Pr(\mathbf{Y} = \mathbf{w})$, *where* $\mathbf{w} \in \mathbb{R}^d$. *Let* $F_1, F_2, \cdots, F_t : \mathbb{R}^d \to \{0, 1\}$ *be* $t$ *random or deterministic functions. Let* $k'$ *be an integer such that:*

$$\sum_{i=1}^{t} F_i(1|\mathbf{w}) \leq k', \forall \mathbf{w} \in \mathbb{R}^d, \tag{19}$$

*where* $F_i(1|\mathbf{w})$ *denotes the probability that* $F_i(\mathbf{w}) = 1$. *Then, we have the following:*

*(1) If* $W = \{\mathbf{w} \in \mathbb{R}^d : Pr(\mathbf{Y} = \mathbf{w})/Pr(\mathbf{X} = \mathbf{w}) \leq \mu\}$ *for some* $\mu > 0$ *and* $\frac{\sum_{i=1}^{t} Pr(F_i(\mathbf{X})=1)}{k'} \geq Pr(\mathbf{X} \in W)$, *then* $\frac{\sum_{i=1}^{t} Pr(F_i(\mathbf{Y})=1)}{k'} \geq Pr(\mathbf{Y} \in W)$.

*(2) If* $W = \{\mathbf{w} \in \mathbb{R}^d : Pr(\mathbf{Y} = \mathbf{w})/Pr(\mathbf{X} = \mathbf{w}) \geq \mu\}$ *for some* $\mu > 0$ *and* $\frac{\sum_{i=1}^{t} Pr(F_i(\mathbf{X})=1)}{k'} \leq Pr(\mathbf{X} \in W)$, *then* $\frac{\sum_{i=1}^{t} Pr(F_i(\mathbf{Y})=1)}{k'} \leq Pr(\mathbf{Y} \in W)$.

*Proof.* We first prove part (1). For convenience, we denote the complement of $W$ as $W^c$. Then, we have the following:

$$\frac{\sum_{i=1}^{t} Pr(F_i(\mathbf{Y}) = 1)}{k'} - Pr(\mathbf{Y} \in W) \tag{20}$$

$$= \int_{\mathbb{R}^d} \frac{\sum_{i=1}^t F_i(1|\mathbf{w})}{k'} \cdot \Pr(\mathbf{Y} = \mathbf{w})d\mathbf{w} - \int_W \Pr(\mathbf{Y} = \mathbf{w})d\mathbf{w} \tag{21}$$

$$= \int_{W^c} \frac{\sum_{i=1}^t F_i(1|\mathbf{w})}{k'} \cdot \Pr(\mathbf{Y} = \mathbf{w})d\mathbf{w} + \int_W \frac{\sum_{i=1}^t F_i(1|\mathbf{w})}{k'} \cdot \Pr(\mathbf{Y} = \mathbf{w})d\mathbf{w} - \int_W \Pr(\mathbf{Y} = \mathbf{w})d\mathbf{w} \tag{22}$$

$$= \int_{W^c} \frac{\sum_{i=1}^t F_i(1|\mathbf{w})}{k'} \cdot \Pr(\mathbf{Y} = \mathbf{w})d\mathbf{w} - \int_W (1 - \frac{\sum_{i=1}^t F_i(1|\mathbf{w})}{k'}) \cdot \Pr(\mathbf{Y} = \mathbf{w})d\mathbf{w} \tag{23}$$

$$\geq \mu \cdot [\int_{W^c} \frac{\sum_{i=1}^t F_i(1|\mathbf{w})}{k'} \cdot \Pr(\mathbf{X} = \mathbf{w})d\mathbf{w} - \int_W (1 - \frac{\sum_{i=1}^t F_i(1|\mathbf{w})}{k'}) \cdot \Pr(\mathbf{X} = \mathbf{w})d\mathbf{w}] \tag{24}$$

$$= \mu \cdot [\int_{W^c} \frac{\sum_{i=1}^t F_i(1|\mathbf{w})}{k'} \cdot \Pr(\mathbf{X} = \mathbf{w})d\mathbf{w} + \int_W \frac{\sum_{i=1}^t F_i(1|\mathbf{w})}{k'} \cdot \Pr(\mathbf{X} = \mathbf{w})d\mathbf{w} - \int_W \Pr(\mathbf{X} = \mathbf{w})d\mathbf{w}] \tag{25}$$

$$= \mu \cdot [\int_{\mathbb{R}^d} \frac{\sum_{i=1}^t F_i(1|\mathbf{w})}{k'} \cdot \Pr(\mathbf{X} = \mathbf{w})d\mathbf{w} - \int_W \Pr(\mathbf{X} = \mathbf{w})d\mathbf{w}] \tag{26}$$

$$= \mu \cdot [\frac{\sum_{i=1}^t \Pr(F_i(\mathbf{X}) = 1)}{k'} - \Pr(\mathbf{X} \in W)] \tag{27}$$

$$\geq 0. \tag{28}$$

We have Equation 24 from 23 due to the fact that $\Pr(\mathbf{Y} = \mathbf{w})/\Pr(\mathbf{X} = \mathbf{w}) \leq \mu, \forall \mathbf{w} \in W$, $\Pr(\mathbf{Y} = \mathbf{w})/\Pr(\mathbf{X} = \mathbf{w}) > \mu, \forall \mathbf{w} \in W^c$, and $1 - \frac{\sum_{i=1}^t F_i(1|\mathbf{w})}{k'} \geq 0$. Similarly, we can prove the part (2). We omit the details for conciseness reason. □

We apply the above lemma to random variables $\mathbf{X}$ and $\mathbf{Y}$, and obtain the following lemma:

**Lemma 4.** *Let* $\mathbf{X} \sim \mathcal{N}(\mathbf{x}, \sigma^2 I)$, $\mathbf{Y} \sim \mathcal{N}(\mathbf{x} + \delta, \sigma^2 I)$, $F_1, F_2, \cdots, F_t : \mathbb{R}^d \to \{0,1\}$ *be* $t$ *random or deterministic functions, and* $k'$ *be an integer such that:*

$$\sum_{i=1}^t F_i(1|\mathbf{w}) \leq k', \forall \mathbf{w} \in \mathbb{R}^d, \tag{29}$$

*where* $F_i(1|\mathbf{w})$ *denote the probability that* $F_i(\mathbf{w}) = 1$. *Then, we have the following:*

*(1) If* $W = \{\mathbf{w} \in \mathbb{R}^d : \delta^T \mathbf{w} \leq \beta\}$ *for some* $\beta$ *and* $\frac{\sum_{i=1}^t Pr(F_i(\mathbf{X})=1)}{k'} \geq Pr(\mathbf{X} \in W)$, *then* $\frac{\sum_{i=1}^t Pr(F_i(\mathbf{Y})=1)}{k'} \geq Pr(\mathbf{Y} \in W)$.

*(2) If* $W = \{\mathbf{w} \in \mathbb{R}^d : \delta^T \mathbf{w} \geq \beta\}$ *for some* $\beta$ *and* $\frac{\sum_{i=1}^t Pr(F_i(\mathbf{X})=1)}{k'} \leq Pr(\mathbf{X} \in W)$, *then* $\frac{\sum_{i=1}^t Pr(F_i(\mathbf{Y})=1)}{k'} \leq Pr(\mathbf{Y} \in W)$.

By leveraging Lemma 2, Lemma 3, and Lemma 4, we derive the following lemma:

**Lemma 5.** *Suppose we have an arbitrary base multi-label classifier* $f$, *an integer* $k'$, *an input* $\mathbf{x}$, *an arbitrary set denoted as* $O$, *two label probability bounds* $\underline{p}_O$ *and* $\overline{p}_O$ *that satisfy* $\underline{p}_O \leq p_O = \sum_{i \in O} Pr(i \in f_{k'}(\mathbf{X})) \leq \overline{p}_O$, *as well as regions* $\mathcal{A}_O$ *and* $\mathcal{B}_O$ *defined as follows:*

$$\mathcal{A}_O = \{\mathbf{w} : \delta^T(\mathbf{w} - \mathbf{x}) \leq \sigma \|\delta\|_2 \Phi^{-1}(\frac{\underline{p}_O}{k'})\} \tag{30}$$

$$\mathcal{B}_O = \{\mathbf{w} : \delta^T(\mathbf{w} - \mathbf{x}) \geq \sigma \|\delta\|_2 \Phi^{-1}(1 - \frac{\overline{p}_O}{k'})\} \tag{31}$$

*Then, we have:*

$$Pr(\mathbf{X} \in \mathcal{A}_O) \leq \frac{\sum_{i \in O} Pr(i \in f_{k'}(\mathbf{X}))}{k'} \leq Pr(\mathbf{X} \in \mathcal{B}_O) \tag{32}$$

$$Pr(\mathbf{Y} \in \mathcal{A}_O) \leq \frac{\sum_{i \in O} Pr(i \in f_{k'}(\mathbf{Y}))}{k'} \leq Pr(\mathbf{Y} \in \mathcal{B}_O) \tag{33}$$

*Proof.* We know $\Pr(\mathbf{X} \in \mathcal{A}_O) = \frac{p_O}{k'}$ based on Lemma 2. Moreover, based on the condition $\underline{p_O} \leq \sum_{i \in O} \Pr(i \in f_{k'}(\mathbf{X}))$, we obtain the first inequality in Equation 32. Similarly, we can obtain the second inequality in Equation 32. We define $F_i(\mathbf{w}) = \mathbb{I}(i \in f_{k'}(\mathbf{w})), \forall i \in O$, where $\mathbb{I}$ is indicator function. Then, we have $\Pr(\mathbf{X} \in \mathcal{A}_O) \leq \frac{\sum_{i \in O} \Pr(i \in f_{k'}(\mathbf{X}))}{k'} = \frac{\sum_{i \in O} \Pr(F_i(\mathbf{X})=1)}{k'}$. Note that there are $k'$ elements in $f_{k'}(\mathbf{w}), \forall \mathbf{w} \in \mathbb{R}^d$, therefore, we have $\sum_{i \in O} F_i(1|\mathbf{w}) = \sum_{i \in O} \mathbb{I}(i \in f_{k'}(\mathbf{w})) \leq k', \forall \mathbf{w} \in \mathbb{R}^d$. Then, we can apply Lemma 4 and we have the following:

$$\Pr(\mathbf{Y} \in \mathcal{A}_O) \leq \frac{\sum_{i \in O} \Pr(F_i(\mathbf{Y}) = 1)}{k'} = \frac{\sum_{i \in O} \Pr(i \in f_{k'}(\mathbf{Y}))}{k'}, \tag{34}$$

which is the first inequality in Equation 33. Similarly, we can obtain the second inequality in Equation 33. □

Based on Lemma 1 and Lemma 2, we derive the following lemma:

**Lemma 6.** *Suppose we have an arbitrary base multi-label classifier $f$, an integer $k'$, an input $\mathbf{x}$, an arbitrary label which is denoted as $l$, two label probability bounds $\underline{p_l}$ and $\overline{p}_l$ that satisfy $\underline{p_l} \leq p_l = Pr(l \in f_{k'}(\mathbf{X})) \leq \overline{p}_l$, and regions $\mathcal{A}_l$ and $\mathcal{B}_l$ defined as follows:*

$$\mathcal{A}_l = \{\mathbf{w} : \delta^T(\mathbf{w} - \mathbf{x}) \leq \sigma \|\delta\|_2 \, \Phi^{-1}(\underline{p_l})\} \tag{35}$$

$$\mathcal{B}_l = \{\mathbf{w} : \delta^T(\mathbf{w} - \mathbf{x}) \geq \sigma \|\delta\|_2 \, \Phi^{-1}(1 - \overline{p}_l)\} \tag{36}$$

*Then, we have:*

$$Pr(\mathbf{X} \in \mathcal{A}_l) \leq Pr(l \in f_{k'}(\mathbf{X})) \leq Pr(\mathbf{X} \in \mathcal{B}_l) \tag{37}$$

$$Pr(\mathbf{Y} \in \mathcal{A}_l) \leq Pr(l \in f_{k'}(\mathbf{Y})) \leq Pr(\mathbf{Y} \in \mathcal{B}_l) \tag{38}$$

*Proof.* We know $\Pr(\mathbf{X} \in \mathcal{A}_l) = \underline{p_l}$ based on Lemma 2. Moreover, based on the condition $\underline{p_l} \leq \Pr(l \in f_{k'}(\mathbf{X}))$, we obtain the first inequality in Equation 37. Similarly, we can obtain the second inequality in Equation 37. We define $F(\mathbf{w}) = \mathbb{I}(l \in f_{k'}(\mathbf{w}))$. Based on the first inequality in Equation 37, we know $\Pr(\mathbf{X} \in \mathcal{A}_l) \leq \Pr(l \in f_{k'}(\mathbf{X})) = \Pr(F(\mathbf{X}) = 1)$. Then, we apply Lemma 1 and we have the following:

$$\Pr(\mathbf{Y} \in \mathcal{A}_l) \leq \Pr(F(\mathbf{Y}) = 1) = \Pr(l \in f_{k'}(\mathbf{Y})), \tag{39}$$

which is the first inequality in Equation 38. The second inequality in Equation 38 can be obtained similarly. □

Next, we formally show our proof for Theorem 1.

*Proof.* We leverage the law of contraposition to prove our theorem. Roughly speaking, if we have a statement: $P \to Q$, then, it's contrapositive is: $\neg Q \to \neg P$, where $\neg$ denotes negation. The law of contraposition claims that a statement is true if, and only if, its contrapositive is true. We define the predicate $P$ as follows:

$$\max\{\Phi(\Phi^{-1}(\underline{p_{a_e}}) - \frac{R}{\sigma}), \max_{u=1}^{d-e+1} \frac{k'}{u} \cdot \Phi(\Phi^{-1}(\frac{p_{\Gamma_u}}{k'}) - \frac{R}{\sigma})\}$$
$$> \min\{\Phi(\Phi^{-1}(\overline{p}_{b_s}) + \frac{R}{\sigma}), \max_{v=1}^{k-e+1} \frac{k'}{v} \cdot \Phi(\Phi^{-1}(\frac{\overline{p}_{\Lambda_v}}{k'}) + \frac{R}{\sigma})\}. \tag{40}$$

We define the predicate $Q$ as follows:

$$\min_{\delta, \|\delta\|_2 \leq R} |L(\mathbf{x}) \cap g_k(\mathbf{x} + \delta)| \geq e. \tag{41}$$

We will first prove the statement: $P \to Q$. To prove it, we consider its contrapositive, i.e., we prove the following statement: $\neg Q \to \neg P$.

**Deriving necessary condition:** Suppose $\neg Q$ is true, i.e., $\min_{\delta, \|\delta\|_2 \leq R} |L(\mathbf{x}) \cap g_k(\mathbf{x} + \delta)| < e$. On the one hand, this means there exist at least $d - e + 1$ elements in $L(\mathbf{x})$ do not appear in $g_k(\mathbf{x} + \delta)$. For convenience, we use $\mathcal{U}_r \subseteq L(\mathbf{x})$ to denote those elements, a subset of $L(\mathbf{x})$ with $r$ elements where $r = d - e + 1$. On the other hand, there exist at least $k - e + 1$ elements in $\{1, 2, \cdots, c\} \setminus L(\mathbf{x})$ appear

in $g_k(\mathbf{x} + \delta)$. We use $\mathcal{V}_s \subseteq \{1, 2, \cdots, c\} \setminus L(\mathbf{x})$ to denote them, a subset of $\{1, 2, \cdots, c\} \setminus L(\mathbf{x})$ with $s = k - e + 1$ elements. Formally, we have the following:

$$\exists\, \mathcal{U}_r \subseteq L(\mathbf{x}), \mathcal{U}_r \cap g_k(\mathbf{x} + \delta) = \emptyset \tag{42}$$

$$\exists\, \mathcal{V}_s \subseteq \{1, 2, \cdots, c\} \setminus L(\mathbf{x}), \mathcal{V}_s \subseteq g_k(\mathbf{x} + \delta), \tag{43}$$

In other words, there exist sets $\mathcal{U}_r$ and $\mathcal{V}_s$ such that the adversarially perturbed label probability $p_i^*$'s for elements in $\mathcal{V}_s$ are no smaller than these for the elements in $\mathcal{U}_r$. Formally, we have the following necessary condition if $|L(\mathbf{x}) \cap g_k(\mathbf{x} + \delta)| < e$:

$$\min_{\mathcal{U}_r} \max_{i \in \mathcal{U}_r} \Pr(i \in f_{k'}(\mathbf{Y})) \leq \max_{\mathcal{V}_s} \min_{j \in \mathcal{V}_s} \Pr(j \in f_{k'}(\mathbf{Y})) \tag{44}$$

**Bounding** $\max_{i \in \mathcal{U}_r} \Pr(i \in f_{k'}(\mathbf{Y}))$ **and** $\min_{j \in \mathcal{V}_s} \Pr(j \in f_{k'}(\mathbf{Y}))$ **for given** $\mathcal{U}_r$ **and** $\mathcal{V}_s$**:** For simplicity, we assume $\mathcal{U}_r = \{w_1, w_2, \cdots, w_r\}$. Without loss of generality, we assume $\underline{p}_{w_1} \geq \underline{p}_{w_2} \geq \cdots \geq \underline{p}_{w_r}$. Similarly, we assume $\mathcal{V}_s = \{z_1, z_2, \cdots, z_s\}$ and $\overline{p}_{z_s} \geq \cdots \geq \overline{p}_{z_2} \geq \overline{p}_{z_1}$. For an arbitrary element $i \in \mathcal{U}_r$, we define the following region:

$$\mathcal{A}_i = \{\mathbf{w} : \delta^T(\mathbf{w} - \mathbf{x}) \leq \sigma \|\delta\|_2 \Phi^{-1}(\underline{p}_i)\} \tag{45}$$

Then, we have the following for any $i \in \mathcal{U}_r$:

$$\Pr(i \in f_{k'}(\mathbf{Y})) \geq \Pr(\mathbf{Y} \in \mathcal{A}_i) = \Phi(\Phi^{-1}(\underline{p}_i) - \frac{\|\delta\|_2}{\sigma}) \tag{46}$$

We obtain the first inequality from Lemma 6, and the second equality from Lemma 2. Similarly, for an arbitrary element $j \in \mathcal{V}_s$, we define the following region:

$$\mathcal{B}_j = \{\mathbf{w} : \delta^T(\mathbf{w} - \mathbf{x}) \geq \sigma \|\delta\|_2 \Phi^{-1}(1 - \overline{p}_j)\} \tag{47}$$

Then, based on Lemma 6 and Lemma 2, we have the following:

$$\Pr(j \in f_{k'}(\mathbf{Y})) \leq \Pr(\mathbf{Y} \in \mathcal{B}_j) = \Phi(\Phi^{-1}(\overline{p}_j) + \frac{\|\delta\|_2}{\sigma}) \tag{48}$$

Therefore, we have the following:

$$\max_{i \in \mathcal{U}_r} \Pr(i \in f_{k'}(\mathbf{Y})) \tag{49}$$

$$\geq \max_{i \in \mathcal{U}_r} \Phi(\Phi^{-1}(\underline{p}_i) - \frac{\|\delta\|_2}{\sigma}) = \max_{i \in \{w_1, w_2, \cdots, w_r\}} \Phi(\Phi^{-1}(\underline{p}_i) - \frac{\|\delta\|_2}{\sigma}) = \Phi(\Phi^{-1}(\underline{p}_{w_1}) - \frac{\|\delta\|_2}{\sigma}) \tag{50}$$

$$\min_{j \in \mathcal{V}_s} \Pr(j \in f_{k'}(\mathbf{Y})) \tag{51}$$

$$\leq \min_{j \in \mathcal{V}_s} \Phi(\Phi^{-1}(\overline{p}_j) + \frac{\|\delta\|_2}{\sigma}) = \min_{j \in \{z_1, z_2, \cdots, z_s\}} \Phi(\Phi^{-1}(\overline{p}_j) + \frac{\|\delta\|_2}{\sigma}) = \Phi(\Phi^{-1}(\overline{p}_{z_1}) + \frac{\|\delta\|_2}{\sigma}) \tag{52}$$

Next, we consider all possible subsets of $\mathcal{U}_r$ and $\mathcal{V}_s$. We denote $\Gamma_u \subseteq \mathcal{U}_r$, a subset of $u$ elements in $\mathcal{U}_r$, and denote $\Lambda_v \subseteq \mathcal{V}_s$, a subset of $v$ elements in $\mathcal{V}_s$. Then, we have the following:

$$\max_{i \in \mathcal{U}_r} \Pr(i \in f_{k'}(\mathbf{Y})) \geq \max_{\Gamma_u \subseteq \mathcal{U}_r} \max_{i \in \Gamma_u} \Pr(i \in f_{k'}(\mathbf{Y})) \tag{53}$$

$$\min_{j \in \mathcal{V}_s} \Pr(j \in f_{k'}(\mathbf{Y})) \leq \min_{\Lambda_v \subseteq \mathcal{V}_s} \min_{j \in \Lambda_v} \Pr(j \in f_{k'}(\mathbf{Y})) \tag{54}$$

We define the following quantities:

$$\underline{p}_{\Gamma_u} = \sum_{i \in \Gamma_u} \underline{p}_i \text{ and } \overline{p}_{\Lambda_v} = \sum_{j \in \Lambda_v} \overline{p}_j \tag{55}$$

Given these quantities, we define the following region based on Equation 30:

$$\mathcal{A}_{\Gamma_u} = \{\mathbf{w} : \delta^T(\mathbf{w} - \mathbf{x}) \leq \sigma \|\delta\|_2 \Phi^{-1}(\frac{\underline{p}_{\Gamma_u}}{k'})\} \tag{56}$$

$$\mathcal{B}_{\Lambda_v} = \{\mathbf{w} : \delta^T(\mathbf{w} - \mathbf{x}) \geq \sigma \|\delta\|_2 \Phi^{-1}(1 - \frac{\overline{p}_{\Lambda_v}}{k'})\} \tag{57}$$

Then, we have the following:

$$\frac{\sum_{i \in \Gamma_u} \Pr(i \in f_{k'}(\mathbf{Y}))}{k'} \tag{58}$$

$$\geq \Pr(\mathbf{Y} \in \mathcal{A}_{\Gamma_u}) \tag{59}$$

$$= \Phi(\Phi^{-1}(\frac{p_{\Gamma_u}}{k'}) - \frac{\|\delta\|_2}{\sigma}) \tag{60}$$

We have Equation 59 from 58 based on Lemma 5, and we have Equation 60 from 59 based on Lemma 2. Therefore, we have the following:

$$\max_{i \in \Gamma_u} \Pr(i \in f_{k'}(\mathbf{Y})) \tag{61}$$

$$\geq \frac{\sum_{i \in \Gamma_u} \Pr(i \in f_{k'}(\mathbf{Y}))}{u} \tag{62}$$

$$= \frac{k'}{u} \cdot \Phi(\Phi^{-1}(\frac{p_{\Gamma_u}}{k'}) - \frac{\|\delta\|_2}{\sigma}) \tag{63}$$

We have Equation 62 from 61 because the maximum value is no smaller than the average value. Similarly, we have the following:

$$\min_{j \in \Lambda_v} \Pr(j \in f_{k'}(\mathbf{Y})) \leq \frac{k'}{v} \cdot \Phi(\Phi^{-1}(\frac{\overline{p}_{\Lambda_v}}{k'}) + \frac{\|\delta\|_2}{\sigma}) \tag{64}$$

Recall that we have $\overline{p}_{w_1} \geq \overline{p}_{w_2} \geq \cdots \geq \overline{p}_{w_r}$ for $\mathcal{U}_r$. By taking all possible $\Gamma_u$ with $u$ elements into consideration, we have the following:

$$\max_{i \in \mathcal{U}_r} \Pr(i \in f_{k'}(\mathbf{Y})) \geq \max_{\Gamma_u \subseteq \mathcal{U}_r} \max_{i \in \Gamma_u} \Pr(i \in f_{k'}(\mathbf{Y})) \geq \max_{\Gamma_u = \{w_1, \cdots, w_u\}} \frac{k'}{u} \cdot \Phi(\Phi^{-1}(\frac{p_{\Gamma_u}}{k'}) - \frac{\|\delta\|_2}{\sigma}) \tag{65}$$

In other words, we only need to consider $\Gamma_u = \{w_1, \cdots, w_u\}$, i.e., a subset of $u$ elements in $\mathcal{U}_r$ whose label probability upper bounds are the largest, where ties are broken uniformly at random. The reason is that $\Phi(\Phi^{-1}(\frac{p_{\Gamma_u}}{k'}) - \frac{\|\delta\|_2}{\sigma})$ increases as $\underline{p_{\Gamma_u}}$ increases. Combining with Equations 49, we have the following:

$$\max_{i \in \mathcal{U}_r} \Pr(i \in f_{k'}(\mathbf{Y})) \geq \max\{\Phi(\Phi^{-1}(\underline{p_{w_1}}) - \frac{\|\delta\|_2}{\sigma}), \max_{\Gamma_u = \{w_1, \cdots, w_u\}} \frac{k'}{u} \cdot \Phi(\Phi^{-1}(\frac{p_{\Gamma_u}}{k'}) - \frac{\|\delta\|_2}{\sigma})\} \tag{66}$$

Similarly, we have the following:

$$\min_{j \in \mathcal{V}_s} \Pr(j \in f_{k'}(\mathbf{Y})) \leq \min\{\Phi(\Phi^{-1}(\overline{p}_{z_1}) + \frac{\|\delta\|_2}{\sigma}), \min_{\Lambda_v = \{z_1, \cdots, z_v\}} \frac{k'}{v} \cdot \Phi(\Phi^{-1}(\frac{\overline{p}_{\Lambda_v}}{k'}) + \frac{\|\delta\|_2}{\sigma})\} \tag{67}$$

**Bounding** $\min_{\mathcal{U}_r} \max_{i \in \mathcal{U}_r} \Pr(i \in f_{k'}(\mathbf{Y}))$ **and** $\max_{\mathcal{V}_s} \min_{j \in \mathcal{V}_s} \Pr(j \in f_{k'}(\mathbf{Y}))$: We have the following:

$$\min_{\mathcal{U}_r} \max_{i \in \mathcal{U}_r} \Pr(i \in f_{k'}(\mathbf{Y})) \tag{68}$$

$$\geq \min_{\mathcal{U}_r} \max\{\max_{i \in \{w_1, w_2, \cdots, w_r\}} \Phi(\Phi^{-1}(\underline{p_i}) - \frac{\|\delta\|_2}{\sigma}), \max_{\Gamma_u = \{w_1, \cdots, w_u\}} \frac{k'}{u} \cdot \Phi(\Phi^{-1}(\frac{p_{\Gamma_u}}{k'}) - \frac{\|\delta\|_2}{\sigma})\} \tag{69}$$

$$\geq \max\{\max_{i \in \{a_e, a_{e+1}, \cdots, a_k\}} \Phi(\Phi^{-1}(\underline{p_i}) - \frac{\|\delta\|_2}{\sigma}), \max_{\Gamma_u = \{a_e, \cdots, a_{e+u-1}\}} \frac{k'}{u} \cdot \Phi(\Phi^{-1}(\frac{p_{\Gamma_u}}{k'}) - \frac{\|\delta\|_2}{\sigma})\} \tag{70}$$

$$= \max\{\Phi(\Phi^{-1}(\underline{p_{a_e}}) - \frac{\|\delta\|_2}{\sigma}), \max_{\Gamma_u = \{a_e, \cdots, a_{e+u-1}\}} \frac{k'}{u} \cdot \Phi(\Phi^{-1}(\frac{p_{\Gamma_u}}{k'}) - \frac{\|\delta\|_2}{\sigma})\} \tag{71}$$

$$= \max\{\Phi(\Phi^{-1}(\underline{p_{a_e}}) - \frac{\|\delta\|_2}{\sigma}), \max_{u=1}^{d-e+1} \frac{k'}{u} \cdot \Phi(\Phi^{-1}(\frac{p_{\Gamma_u}}{k'}) - \frac{\|\delta\|_2}{\sigma})\}, \tag{72}$$

---

**Algorithm 1:** Computing the Certified Intersection Size

---

**Input:** $f$, $\mathbf{x}$, $L(\mathbf{x})$, $R$, $k'$, $k$, $n$, $\sigma$, and $\alpha$.
**Output:** Certified intersection size.
$\mathbf{x}^1, \mathbf{x}^2, \cdots, \mathbf{x}^n \leftarrow \text{RANDOMSAMPLE}(\mathbf{x}, \sigma)$
$\text{counts}[i] \leftarrow \sum_{t=1}^{n} \mathbb{I}(i \in f(\mathbf{x}^t)), i = 1, 2, \cdots, c.$
$\underline{p_i}, \overline{p}_j \leftarrow \text{PROBBOUNDESTIMATION}(\text{counts}, \alpha), i \in L(\mathbf{x}), j \in \{1, 2, \cdots, c\} \setminus L(\mathbf{x})$
$e \leftarrow \text{BINARYSEARCH}(\sigma, k', k, R, \{\underline{p_i} | i \in L(\mathbf{x})\}, \{\overline{p}_j | j \in \{1, 2, \cdots, c\} \setminus L(\mathbf{x})\})$
**return** $e$

---

where $\Gamma_u = \{a_e, \cdots, a_{e+u-1}\}$. We have Equation 70 from 69 because $\max\{\max_{i \in \{w_1, w_2, \cdots, w_r\}} \Phi(\Phi^{-1}(\underline{p_i}) - \frac{\|\delta\|_2}{\sigma}), \max_{\Gamma_u = \{w_1, \cdots, w_u\}} \frac{k'}{u} \cdot \Phi(\Phi^{-1}(\frac{p_{\Gamma_u}}{k'}) - \frac{\|\delta\|_2}{\sigma})\}$ reaches the minimal value when $\mathcal{U}_r$ contains $r$ elements with smallest label probability lower bounds, i.e., $\mathcal{U}_r = \{a_e, a_{e+1}, \cdots, a_d\}$, where $r = d - e + 1$. Similarly, we have the following:

$$\max_{\mathcal{V}_s} \min_{j \in \mathcal{V}_s} \text{Pr}(j \in f_{k'}(\mathbf{Y})) \leq \min\{\Phi(\Phi^{-1}(\underline{p_{b_s}}) + \frac{\|\delta\|_2}{\sigma}), \min_{v=1}^{s} \frac{k'}{v} \cdot \Phi(\Phi^{-1}(\frac{\overline{p}_{\Lambda_v}}{k'}) + \frac{\|\delta\|_2}{\sigma})\}, \quad (73)$$

where $\Lambda_v = \{b_{s-v+1}, \cdots, b_s\}$ and $s = k - e + 1$.

**Applying the law of contraposition:** Based on necessary condition in Equation 44, if we have $|T \cap g_k(\mathbf{x} + \delta)| < e$, then, we must have the following:

$$\max\{\Phi(\Phi^{-1}(\underline{p_{a_e}}) - \frac{\|\delta\|_2}{\sigma}), \max_{u=1}^{d-e+1} \frac{k'}{u} \cdot \Phi(\Phi^{-1}(\frac{p_{\Gamma_u}}{k'}) - \frac{\|\delta\|_2}{\sigma})\} \tag{74}$$

$$\leq \min_{\mathcal{U}_r} \max_{i \in \mathcal{U}_r} \text{Pr}(i \in f_{k'}(\mathbf{Y})) \tag{75}$$

$$\leq \max_{\mathcal{V}_s} \min_{j \in \mathcal{V}_s} \text{Pr}(j \in f_{k'}(\mathbf{Y})) \tag{76}$$

$$\leq \min\{\Phi(\Phi^{-1}(\overline{p}_{b_e}) + \frac{\|\delta\|_2}{\sigma}), \min_{v=1}^{k-e+1} \frac{k'}{v} \cdot \Phi(\Phi^{-1}(\frac{\overline{p}_{\Lambda_v}}{k'}) + \frac{\|\delta\|_2}{\sigma})\}, \tag{77}$$

We apply the law of contraposition and we obtain the statement: if we have the following:

$$\max\{\Phi(\Phi^{-1}(\underline{p_{a_e}}) - \frac{\|\delta\|_2}{\sigma}), \max_{u=1}^{d-e+1} \frac{k'}{u} \cdot \Phi(\Phi^{-1}(\frac{p_{\Gamma_u}}{k'}) - \frac{\|\delta\|_2}{\sigma})\}$$

$$> \min\{\Phi(\Phi^{-1}(\overline{p}_{b_s}) + \frac{\|\delta\|_2}{\sigma}), \max_{v=1}^{k-e+1} \frac{k'}{v} \cdot \Phi(\Phi^{-1}(\frac{\overline{p}_{\Lambda_v}}{k'}) + \frac{\|\delta\|_2}{\sigma})\}, \tag{78}$$

Then, we must have $|L(\mathbf{x}) \cap g_k(\mathbf{x} + \delta)| \geq e$. From Equation 8, we know that Equation 78 is satisfied for $\forall \|\delta\|_2 \leq R$. Therefore, we reach our conclusion. $\qquad \square$

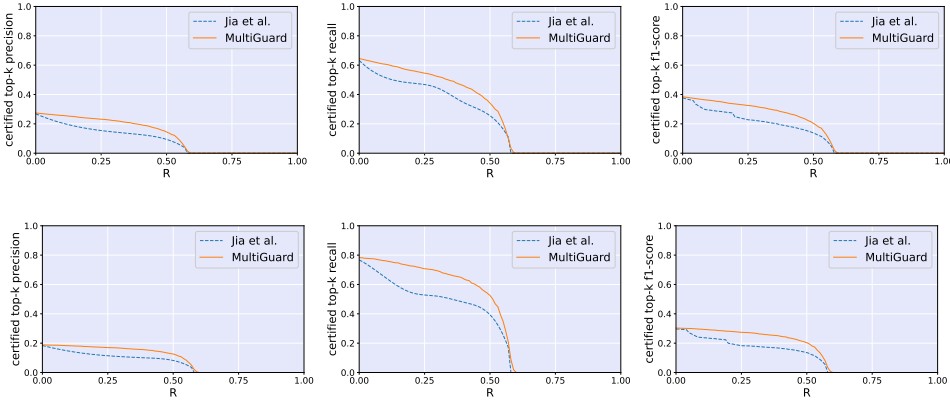

(a) Certified top-$k$ precision@$R$    (b) Certified top-$k$ recall@$R$    (c) Certified top-$k$ f1-score@$R$

**Figure 2: Comparing MultiGuard with with Jia et al. [22] on MS-COCO (first row) and NUS-WIDE (second row) dataset.**

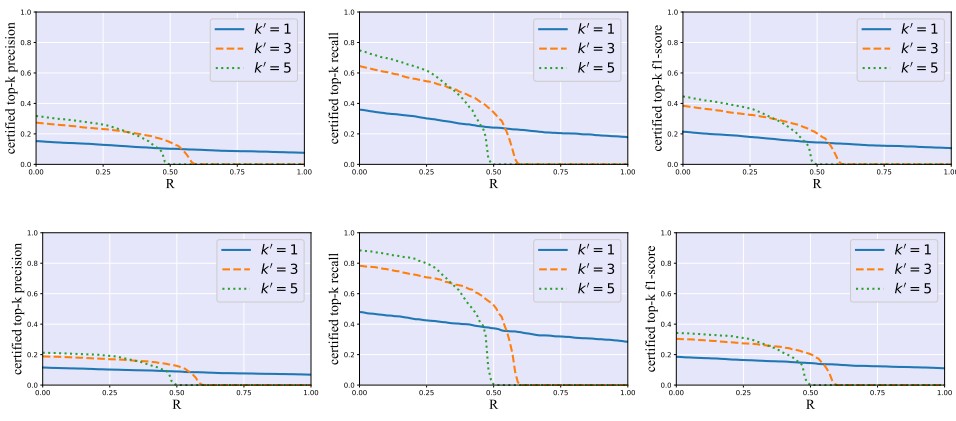

(a) Certified top-$k$ precision@$R$    (b) Certified top-$k$ recall@$R$    (c) Certified top-$k$ f1-score@$R$

**Figure 3: Impact of $k'$ on the certified top-$k$ precision@$R$, certified top-$k$ recall@$R$, and certified top-$k$ f1-score@$R$ on MS-COCO (first row) and NUS-WIDE (second row) dataset.**

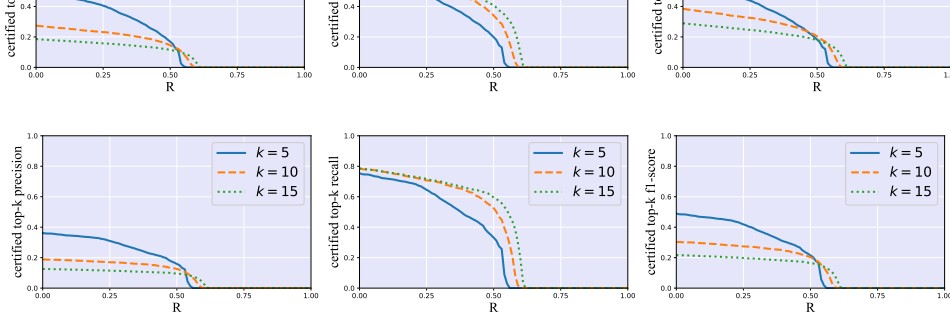

(a) Certified top-$k$ precision@$R$    (b) Certified top-$k$ recall@$R$    (c) Certified top-$k$ f1-score@$R$

**Figure 4: Impact of $k$ on the certified top-$k$ precision@$R$, certified top-$k$ recall@$R$, and certified top-$k$ f1-score@$R$ on MS-COCO (first row) and NUS-WIDE (second row) dataset.**

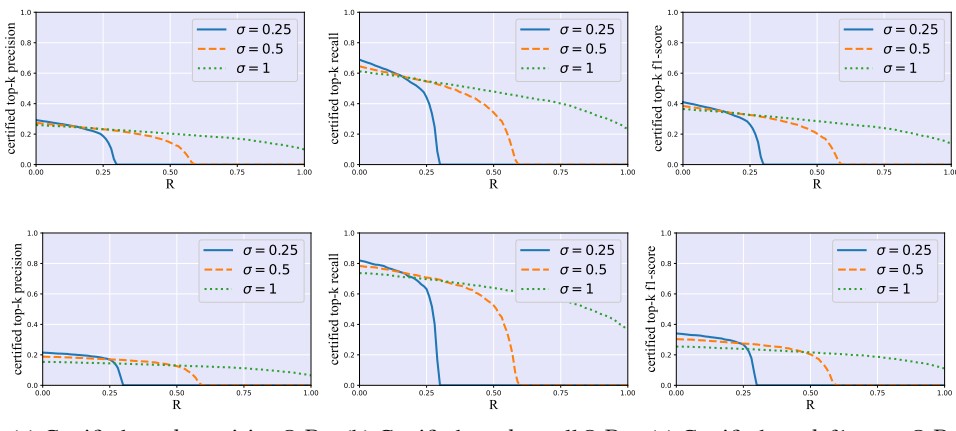

(a) Certified top-$k$ precision@$R$     (b) Certified top-$k$ recall@$R$     (c) Certified top-$k$ f1-score@$R$

**Figure 5: Impact of $\sigma$ on the certified top-$k$ precision@$R$, certified top-$k$ recall@$R$, and certified top-$k$ f1-score@$R$ on MS-COCO (first row) and NUS-WIDE (second row) dataset.**

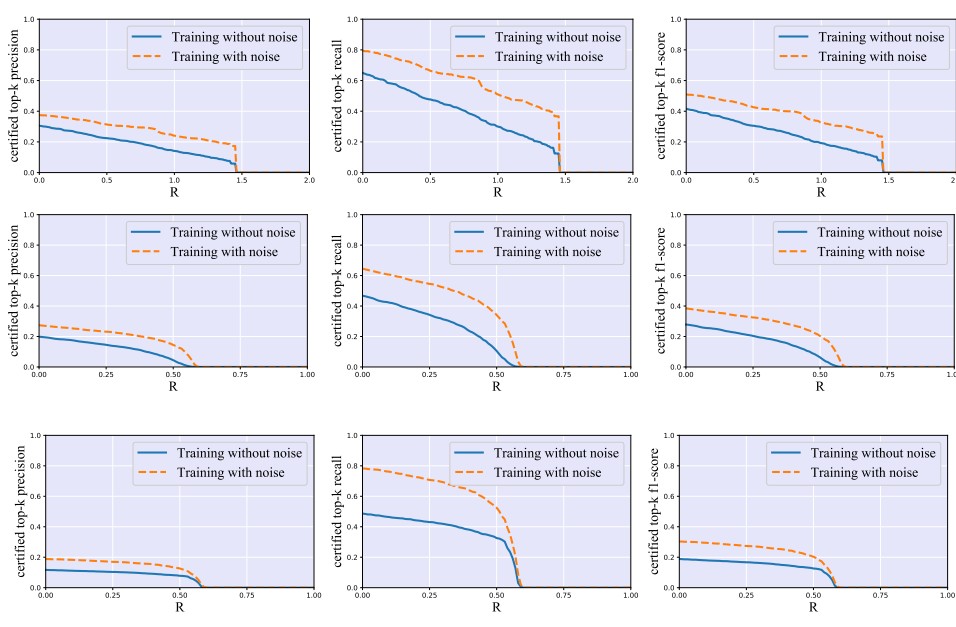

(a) Certified top-$k$ precision@$R$     (b) Certified top-$k$ recall@$R$     (c) Certified top-$k$ f1-score@$R$

**Figure 6: Training the base multi-label classifier with vs. without noise on Pascal VOC (first row), MS-COCO (second row) and NUS-WIDE (third row) datasets.**