# OpenReview forum: "MultiGuard: Provably Robust Multi-label Classification against Adversarial Examples"
_NeurIPS.cc/2022/Conference — NeurIPS 2022 Accept_

### Official Review · Reviewer_RjyH · 2022-07-09

**Rating:** 6
**Confidence:** 3
**Soundness:** 3 good
**Presentation:** 3 good
**Contribution:** 2 fair

**Summary:**

In this work, a multi-label variant of the random smoothing technique is designed by extending the Neyman-Pearson Lemma from the single-output model to the multi-output model. The theoretical study shows that the proposed multi-label random smoothing method can be considered as a more general framework of provably robust learning. It takes the single-label random smoothing work published in [12] as a special variant. Furthermore, considering only one of the multiple labels as the protected target by the injected random smoothing noise, the proposed method can be also reduced to [22].

**Questions:**

The major question is it is unclear whether the proposed theoretical study based on the extended Neyman-Pearson Lemma considers the label correlation explicitly in protecting multi-label systems against adversarial attacks. Furthermore, an injected random smoothing noise may protect some labels, while harm the classification performance of the other labels severely. Could the proposed theoretical analysis provide some insights on how to control the unexpected classification performance drops in multi-label systems ?  This is also the unique problem in robustifying multi-label learning tasks, in contrast to the single-label tasks.

######

We are satisfied with the clarification provided by the author. We think this concern has been well addressed.

**Ethics Review Area:**

["I don’t know"]

**Limitations:**

Please find our concerns in the listed weak points.

**Strengths And Weaknesses:**

Strong points:

1. How to design the random smoothing noise for multi-output classification algorithms is important and remains open to address. It is good to see research efforts devoted to this topic. Extending the Neyman-Pearson Lemma is a decent idea along this track.

2. The theoretical study is solid and points out clearly the relation between the proposed method and the previous random-smoothing techniques.

Weak points:

The core idea of mult-label classification is to make use of the correlation between multiple labels, to boost the learning performance. Highly correlated labels are likely to have similar classification accuracy levels (they tend to be correctly classified / misclassified at the same time). It is unclear how the proposed method considers the label correlation in the design of the random smoothing noise. For example, the provably robustness radius / bound should be similar for highly correlated labels, as the same smoothing perturbation tends to bring similar effects to the correlated label pairs.  Therefore considering untargeted multi-label attacks, the design of the random smoothing noise should be able to benefit from the label correlation:  we can minimise the variance of the random smoothing noise, while providing the provably defense to as many as possible correlated labels.

---

> ### Author Response · Authors · 2022-07-29
> **Response to Reviewer RjyH**
>
> Thanks for the constructive comments!
>
> When leveraging our extended Neyman-Pearson Lemma to derive the certified intersection size, we jointly consider all ground truth labels. As a result, the computation of certified intersection size involves the label probabilities of all ground truth labels as shown in Equation (8). In contrast, Jia et al. consider each ground truth label independently, which achieves a sub-optimal certified robustness guarantee as demonstrated in our experiments.
>
> The standard deviation $\sigma$ of isotropic Gaussian noise (added to a testing input when building our smoothed classifier) controls a tradeoff between robustness and accuracy. In particular, a smaller $\sigma$ can achieve better classification accuracy but is less robust against adversarial examples as shown in our experimental results (fourth row in Figure 1 and Figure 5 in Appendix).

---

> > ### Comment · Reviewer_RjyH · 2022-08-07
> > **Thanks for the response**
> >
> > Thanks for the clarification. I think it addresses my concern well. I will change my rating accordingly. In parallel, we'd encourage the author to integrate the clarification into the paper. This will help readers understand better the value of the contribution.

---

> > > ### Author Response · Authors · 2022-08-07
> > > **Thanks for the comment**
> > >
> > > Many thanks for the comment! We really appreciate the constructive feedback, which significantly improves the paper. We will definitively integrate our clarifications into the paper. Thanks for updating the rating score!

---

### Official Review · Reviewer_9w49 · 2022-07-11

**Rating:** 6
**Confidence:** 2
**Soundness:** 3 good
**Presentation:** 3 good
**Contribution:** 3 good

**Summary:**

This paper discusses the adversarial robustness of multi-label classification task, extend provably robust defense methods to multi-label case, and derive a way to estimate the certified intersection size, which is the least number of true labels in the set of labels predicted by the certified classifier. Specifically, the author extends works of randomized smoothing to multi-label task by leveraging the law of contraposition and a variant of Neyman-Pearson lemma to derive the conditions of robust certification and using MC algorithm to help estimate the certified intersection size.

Experiments on three datasets (VOC 2007, MS-COCO, NUS-WIDE) show that the proposed method performs better than directly extending randomized smoothing (Jia et al.) to multi-label classification. They also do experiments to study the effects of important hyper-parameters.


**Questions:**

NA

**Limitations:**

Though the method performs better than directly extending randomized smoothing on multi-class classification to multi-label case, the performance is far from perfect. There's still room for improvement. The author points out some improvement direction at the end of the work.

**Strengths And Weaknesses:**

Strengths:

- The work is novel as it is the first one to extend provably robust defense to multi-label classification and the extension is not trivial as multi-label classification is different from multi-class classification. Besides, experiments show that directly extending the method proposed for multi-class classification to multi-label does not work well. The authors also show that randomized smoothing methods proposed for multi-class classification can be viewed as special cases of their framework.

- The paper is clearly written and easy to follow. The method is straightforward and not hard to implement. The proposed method works well on three benchmark datasets.

Weakness:

- Though pseudo code and some implementation details are provided, actual code is not provided,  It is better if the actual code can be provided for reproducing the results.

---

> ### Author Response · Authors · 2022-07-29
> **Response to Reviewer 9w49**
>
> Thanks for the constructive comments!
>
> Thanks for the suggestion. Our code can be found in this anonymized address: https://github.com/RandomizedOS/MultiGuard. We are still cleaning code for other experimental results. We will add more.

---

> > ### Comment · Reviewer_9w49 · 2022-08-09
> > **Thanks for the response**
> >
> > Thanks the authors for providing the code. My concerns are addressed.

---

> > > ### Author Response · Authors · 2022-08-09
> > > **Thanks for the note**
> > >
> > > Thanks for your time. We really appreciate the suggestion.

---

### Official Review · Reviewer_95yU · 2022-07-13

**Rating:** 6
**Confidence:** 3
**Soundness:** 3 good
**Presentation:** 4 excellent
**Contribution:** 3 good

**Summary:**

This paper proposes MultiGuard, where multi-label classification with provable guarantees against adversarial perstubations is studied. The method is based on randomized-smoothing, where randomization with Gaussian noise is utilized to provide a smoothed classifier with provable guarantees, and this work generalizes that to multi-label classification, with adjusted claims to suit multi-label classification.

**Questions:**

In order to make the paper self-contained, I’d recommend to add a section on the training procedure of the proposed method. Even though references to ASL for training the base multi-label classification training is given, the paper is not self contained as no further information, i.e. the definition of the loss and training procedure and/or hyper parameters, is given, which makes the work not self-contained.

**Limitations:**

The bounds drop quickly as the multi-label parameter k drops for 1 to higher values (2 or 3), which poses the question of whether these bounds are non-trivial or useful in practice.

**Strengths And Weaknesses:**

This work uses simple yet intuitive tools and techniques, namely a variant of Neyman-Pearson lemma as well as law of contraposition to extend the provable guarantees of randomized-smoothing multi-class classification to that of mutli-label. Eventhough technical novelty is somewhat limited, the problem is of interest and together with the provable performance, the community would benefit from the publication of the work.

Furthermore, the paper is written well and is easy to follow and has good structure. Numerical results are well presented and parameter sensitivity is studied, however in terms of comparison with other methods are somewhat limited, since the method is only compared with Jia et al, and thus could benefit from more extensive comparisons.

---

> ### Author Response · Authors · 2022-07-29
> **Response to Reviewer 95yU**
>
> Thanks for the constructive comments!
>
> We only compare with Jia et al. because it is the only certified defense that considers top-$k$ predictions against $\ell_2$ adversarial perturbation. We will clarify.
>
> Suppose $p_i$ is the probability that a base multi-label classifier predicts label $i (i=1,2,\cdots, c)$ for a training input. Moreover, we let $y_i$ be 1 (or 0) if the label $i$ is (or is not) a ground truth label of the training input. The loss of ASL [2] is as follows:  $L_{ASL}=\sum_{i=1}^{c} -y_{i} L_{i+}-(1-y_{i}) L_{i-}$, where $L_{i+}=(1-p_{i})^{\gamma_{+}} \log (p_{i})$ and $L_{i-}=\left(\max(p_{i}-m,0)\right)^{\gamma_{-}} \log \left(1-\max(p_{i}-m,0)\right)$. Note that $\gamma_{+}$, $\gamma_{-}$, and $m$ are hyperparameters. Following [2], we set $\gamma_{+}=0, \gamma_{-}=4, m=0.05$. We train a base multi-label classifier using Adam optimizer, where the learning rate is $10^{-3}$ and batch size is 32. We adopt the official implementation of ASL (https://github.com/Alibaba-MIIL/ASL) in our experiments. We will add those details to our paper as suggested.
>
> We note that $k’$ achieves a tradeoff between the certified top-$k$ precision@R (or certified top-$k$ recall@R or certified top-$k$ f1-score@R) without attacks and robustness. In practice, we can set $k’=1$ if robustness is desired.

---

### Official Review · Reviewer_7aeN · 2022-07-18

**Rating:** 6
**Confidence:** 5
**Soundness:** 3 good
**Presentation:** 4 excellent
**Contribution:** 3 good

**Summary:**

This paper extends the traditional randomized smoothing, which only supports single-label classifiers, to support multi-label classifiers. The extension is based on a generalization of Neyman-Pearson lemma: the Neyman-Pearson lemma still holds for a normalized sum of functions as long as the normalized sum is within [0, 1] universally. The proposed method is evaluated on a few classical multi-class datasets including VOC 2007, MS-COCO, and NUS-WIDE, and achieves superior performance compared to a naive extension from top-k certification in terms of multi-class precision, recall, and F1 score.

**Questions:**

- In Eqn. (8), where do we constrain the validness of input to $\Phi^{-1}$? In other words, $\dfrac{\underline{p_{A_u}}}{k'}$ or $\dfrac{\overline{p}_{B_v}}{k'}$ seems to be possible to grow larger than 1, is it the case? If so, we may need to fix Eqn. (8) and the corresponding proofs, especially in Eqns. (72) and (78).

**Limitations:**

- After reading the submission, I feel the main technical idea is to use (e+1)-th ground-truth label's lower probability bound to compare with the (e+1)-th adversarial label's upper probability bound to derive the certification. This idea seems to be a bit straightforward. On the other hand, the technique of bagging multiple labels together for certification (second terms in Eqn. (8) constraints) sounds interesting to me. I am wondering whether this bagging technical contributes much to the final certification tightness, or is just incremental. Maybe an ablation study can be conducted to quality this. If the technique contributes much, I would recommend the authors squeeze some space in Section 3.3 (seems most sampling techniques are already in the literature) and expand the discussion of this technique more to highlight the novelty and technical contributions.

- Following the above comments, the experimental section seems to mainly convey "the proposed method is superior than the baseline" but lacks the discussion on **why** it surpasses the baseline. The discussion between Line 360 and 362 looks a bit thin to me.

**Strengths And Weaknesses:**

Strengths:
- Non-trivial technical contributions: a generalization version of Neyman-Pearson lemma and the resulting certification protocol based on the extended randomized smoothing for multi-class certification.
- Handles an important problem: multi-class certification is a common learning task formulation, e.g., in computer vision.
- Significant experimental results: consistently beating an adapted baseline (from top-k randomized smoothing certification).
- Writing quality is good.

Weaknesses:
- Some mathematical illustrations may not be rigor enough. See "Questions" for more details.
- The underlying technical idea is a bit straightforward, or not explained clearly enough in terms of challenges. See "Limitations" for more detail.

Note: I am not familiar with multi-label classification literature. Therefore, I didn't evaluate the corresponding experimental protocols, e.g., the selection of evaluation datasets. I will take other reviewers' comments into account for this part.

---

> ### Author Response · Authors · 2022-07-29
> **Response to Reviewer 7aeN**
>
> Thanks for the constructive comments!
>
> Thanks a lot for pointing it out. $\underline{p_{A_u}}$ is a lower bound of label probability. Thus, we have $\underline{p_{A_u}} \leq p_{A_u} \leq k’$, where the last inequality is based on the fact that $p_{A_u} \leq \sum_{i=1}^{d} p_{a_i} \leq \sum_{i=1}^{c} p_i =k’$. We note that  $\overline{p}\_{B\_{v}}$ is an upper bound of $p\_{B\_{v}}$ which is the summation of the label probabilities of a subset labels among $\{1,2,\cdots,c\}$. Based on the fact that $\sum_{i=1}^{c} p_i =k’$, $k’$ can be viewed as an upper bound of $p\_{B\_{v}}$. In Section 3.3, our estimated $\overline{p}_{B\_{v}}$ will always be no larger than $k’$ (see Line 293), and thus we can apply our Theorem 1. We will clarify.
>
> We perform experiments under our default setting to validate the effectiveness of our bagging terms (i.e., second terms in Eqn. (8)). Our results are as follows: with and without the second terms, MultiGuard respectively achieves 31.3% and  23.6% certified top-k precision@R, 66.4% and 48.8% certified top-k recall@R, as well as 42.6% and 31.8% certified top-k f1-score@R, where the perturbation size  $R = 0.5$ and the dataset is VOC 2007. As the result shows, our second terms can significantly improve certified intersection size. We will add more technical details as suggested.
>
> The reason why MultiGuard is better than Jia et al. is as follows. Given a perturbation size, Jia et al. can verify whether a ground truth label is among the top-$k$ labels predicted by the smoothed classifier by leveraging standard Neyman Pearson Lemma. However, each ground truth label is independently considered by Jia et al., which is sub-optimal. For instance, suppose we have two ground truth labels,  it is very likely that both of them are not in the top-$k$ predicted labels when considered independently, but at least one of them will be among the top-$k$ predicted labels when jointly considered. The intuition is that it is easier for an attacker to find an adversarial perturbation such that a certain label is not in the top-$k$ predicted labels, but it is more challenging for an attacker to find an adversarial perturbation such that both of the two labels are not in the top-$k$ predicted labels. Our MultiGuard can jointly consider all ground truth labels when deriving the certified intersection size and thus is better than Jia et al.. We will add the discussion to our paper.

---

> > ### Comment · Reviewer_7aeN · 2022-08-03
> > **Thanks for the response**
> >
> > I appreciate the authors for your timely and detailed response, and I hope the authors can include these nice clarifications into their paper revision. Most of my concerns are resolved except one minor question:
> >
> > When $k'=1, k\ge 1$ and $|L(x)|=1$, the author mentioned that their certification reduces to Jia et al. Is it because the second terms in Eqn. (8) $\max$ and $\min$ operators become equivalent to the first terms respectively?

---

> > > ### Author Response · Authors · 2022-08-04
> > > **Thanks for the comment**
> > >
> > > Thank you so much for your time and insightful feedback. We will definitively incorporate them into our paper.
> > >
> > > For the max operator, the second term in Eqn. (8) reduces to the first term under those conditions. Moreover, it is equivalent to the term on the left-hand side of Eqn. (3) in Jia et al..  For the min operator, the second term becomes $\min_{v=1}^{s}\frac{1}{v} \cdot\Phi(\Phi^{-1}(\overline{p}\_{B\_{v}})+\frac{R}{\sigma})$ under those conditions. When $v=1$, $\Phi(\Phi^{-1}(\overline{p}\_{B\_{v}})+\frac{R}{\sigma})$ is equivalent to the first term. In other words, the first term is already included by the second term due to its min operation. Moreover, the second term is equivalent to the term on the right-hand side of Eqn. (3) in Jia et al.. Our certification reduces to Jia et al. as both sides are equivalent under those conditions.

---

> > > > ### Comment · Reviewer_7aeN · 2022-08-06
> > > > **Follow-up response**
> > > >
> > > > Thanks for your timely response. My question is addressed. Please make sure to incorporate these illustrations in the updated version of the manuscript. I increased the confidence score of my evaluation accordingly.

---

> > > > > ### Author Response · Authors · 2022-08-07
> > > > > **Thanks for the note**
> > > > >
> > > > > Thanks a lot for the comment! It is our great pleasure to answer the questions. We will definitively add those illustrations in the next version. Thanks for updating the confidence score!

---

### Meta-Review · Area_Chair_4cYv · 2022-08-24

**Recommendation:** Accept
**Confidence:** Certain

**Metareview:**

This paper studies adversarial examples for varieties of randomized smoothing, namely, ways to improve the robustness of a classifier by adding noise and averaging over inputs. The main contribution is MultiGuard, which is a provably robust defense for multi-label classification. Moreover, the method works for a variety of classifiers, and the authors also provide theoretical and empirical results to back up their method.

The reviewers generally find the technical contribution to be significant (although perhaps elementary) as well as finding the problem domain to be important and interesting. The reviewers also found the mathematical tools to be intuitive and appropriate (e.g., a variant of Neyman-Pearson lemma as well as a law of contraposition to extend the provable guarantees of randomized-smoothing multi-class classification to that of mutli-label). In addition to acknowledging the theoretical results, the reviewers also felt that the empirical studies were sufficient to verify the authors’ main findings.

On the negative side, there are some concerns about clarity and rigor of the results. I would encourage the authors to improve the exposition and the preliminaries to increase the readability of the work. Similarly, there are some questions about comparison to prior work and similar ideas that should be addressed.

Overall, I recommend acceptance. The positives outweigh the negatives, and the author-review discussion seemed to address many of the main questions.

**Award:**

No

---

### Decision · Program_Chairs · 2022-09-14

Accept